# OpenTSLM: Time-Series Language Models for Reasoning over Multivariate Medical Text- and Time-Series Data

**Patrick Langer** [1 2 3]  **Thomas Kaar** [1 3]  **Max Rosenblattl** [1 3]  **Maxwell A. Xu** [4 5]  **Winnie Chow** [6]  **Martin Maritsch** [7]
**Robert Jakob** [3]  **Ning Wang** [3]  **Juncheng Liu** [8 9]  **Aradhana Verma** [10]  **Brian Han** [11]  **Daniel Seung Kim** [12]
**Henry Chubb** [11]  **Scott Ceresnak** [11]  **Aydin Zahedivash** [1]  **Alexander Tarlochan Singh Sandhu** [10]
**Fatima Rodriguez** [10]  **Daniel McDuff** [5 13]  **Elgar Fleisch** [2 3 14]  **Oliver Aalami** [1]  **Filipe Barata** [2 *]
**Paul Schmiedmayer** [1 *]

## Abstract

Large language models (LLMs) have shown strong capabilities in interpreting multimodal data but remain limited in handling time-series data. Addressing this gap could help to translate longitudinal and wearable data into actionable insights and patient-facing digital health applications. We propose OpenTSLM, an open-source family of Time-Series Language Models (TSLMs) integrating time-series as a native modality into pre-trained LLMs, enabling natural-language prompting and reasoning over multiple time-series via either soft prompting (OpenTSLM-SoftPrompt) or cross-attention (OpenTSLM-Flamingo). To enable training models for time-series reasoning, we introduce three datasets: HAR-CoT (human activity recognition), Sleep-CoT (sleep staging), and ECG-QA-CoT (electrocardiogram question answering). Across tasks, OpenTSLM models outperform baselines, reaching F1 scores of 69.88% in sleep staging and 67.64% in HAR; OpenTSLM-Flamingo also scales more efficiently in memory as the number and length of time series increase. Expert evaluations with cardiologists show that OpenTSLMs exhibit strong reasoning capabilities on raw ECG data.

*Equal contribution  [1]Stanford University, USA [2]Centre for Digital Health Interventions, ETH Zurich, Switzerland [3]Agentic Systems Lab, ETH Zurich, Switzerland [4]University of Illinois Urbana-Champaign, USA [5]Google Research, USA [6]Stanford University, USA [7]Amazon, USA [8]Microsoft [9]National University of Singapore, Singapore [10]Division of Cardiovascular Medicine, Stanford University, USA [11]Pediatric Cardiology, Stanford University, USA [12]Division of Cardiology, University of Washington, USA [13]University of Washington, USA [14]Centre for Digital Health Interventions, University of St. Gallen, Switzerland. Correspondence to: Patrick Langer <planger@stanford.edu>.

*Proceedings of the 43$^{rd}$ International Conference on Machine Learning*, Seoul, South Korea. PMLR 306, 2026. Copyright 2026 by the author(s).

## 1. Introduction

Medicine is inherently temporal: assessment, diagnosis, and treatment depend on how signs, symptoms, and biomarkers evolve over time (Giannoula et al., 2018; Henly et al., 2011; Jørgensen et al., 2024). Clinical decision-making relies on temporal patterns—tracking vital signs, medication responses, laboratory values, and disease progression markers to guide diagnosis, prognosis, and therapeutic interventions. As time-series data from electronic health records and continuous monitoring proliferate (Abernethy et al., 2022; Marra et al., 2024; Yeung et al., 2023), human-legible representations become essential for interpreting and managing this information (Olex & Mcinnes, 2021; Senathirajah et al., 2020; Zhou et al., 2008). Clinical summaries must translate complex temporal patterns, e.g., biomarker trajectories, into interpretable assessments that support evidence-based decision-making and care coordination.

Recent advances in multimodal large language models (LLMs) allow users to interpret complex data through natural language, synthesizing information across text, images, audio, and video (Wu et al., 2023; AlSaad et al., 2024). However, reasoning over time-series data remains a critical blind spot among currently supported modalities. Prior work has attempted to integrate time-series as plain text tokens (Gruver et al., 2023; Kim et al., 2024; Liu et al., 2023); however results have been limited (Merrill et al., 2024). Other approaches reprogram LLMs to act as feature extractors for classification heads, which then output a fixed set of classes or values, thereby losing text-generation capabilities (Li et al., 2025; Nie et al., 2023; Pillai et al., 2025; Ye et al., 2025). More recently, soft prompting has been explored, concatenating learnable time-series tokens with text tokens to preserve generation (Chow et al., 2024). Yet, longer series may require more tokens, increasing context length (Götz et al., 2025; Nie et al., 2023) and compute due to the quadratic cost of self-attention (Nie et al., 2023; Vaswani et al., 2017).

To overcome prior limitations, we propose Time-Series Lan-

guage Models (TSLMs), which integrate time-series as a native modality in LLMs. TSLMs provide a natural interface to complex medical data, enabling clinicians and patients to query, interpret, and reason about longitudinal health information directly through natural language. We introduce OpenTSLM, a family of TSLMs built by extending pretrained LLMs with time-series inputs. A central design question in building TSLMs is how to represent time-series signals. Prior work has primarily used soft prompting, encoding time-series as learned token embeddings concatenated with text tokens. While lightweight, this captures temporal dependencies only implicitly, as additional tokens in the context, and may scale poorly to longer or multiple sequences. We hypothesize that explicit multimodal fusion via cross-attention may be more effective for modeling temporal structure. To compare both approaches, we explore two variants for OpenTSLM. The first, **OpenTSLM-SoftPrompt** leverages soft prompting to concatenate text tokens with time-series tokens predicted from a learnable encoder, so the model processes both as a single sequence without distinguishing between them. The second, **OpenTSLM-Flamingo**, by contrast, models time-series explicitly as a separate modality, using a cross-attention mechanism inspired by Flamingo (Alayrac et al., 2022) to fuse time-series and text. We construct OpenTSLM-SoftPrompt and OpenTSLM-Flamingo using Llama (Touvron et al., 2023) and Gemma (GemmaTeam et al., 2024) backbones, and pair them with time-series encoders trained from scratch as well as with time-series foundation models (TSFMs) such as Chronos-2 (Ansari et al., 2025). We benchmark these models against each other and against baselines including LLMs with tokenized time-series inputs (Gruver et al., 2023), fine-tuned tokenized time-series models, and vision-based approaches. Unlike prior classification-based approaches, our models are trained in text-based reasoning tasks, generating chain of thought (CoT) rationales before producing predictions. For training and evaluation, we introduce three new datasets: **HAR-CoT**, **Sleep-CoT**, and **ECG-QA-CoT**. To foster reproducibility and further research on TSLMs, we release OpenTSLM as an open-source framework, including models and datasets.[1].

## 2. Related Work

Creating TSLMs remains an open research challenge. The main barrier is the modality gap between continuous signals and discrete text representations (Chow et al., 2024; Pillai et al., 2025; Zhang et al., 2025). Prior work has proposed three main strategies to bridge this gap, as summarized by Zhang et al. (2024): tokenizing time-series as text (Section 2.1), applying soft prompting (Section 2.2), and using cross-attention mechanisms (Section 2.3). Table 1

---

[1] https://github.com/StanfordBDHG/OpenTSLM

provides an overview of relevant methods. For a discussion of OpenTSLM in comparison to prior work, we refer to section 5.

*Table 1.* Methods combining time-series data with LLMs.

| Name | Method | Task | Text Gen. | Multi-Sensor | Raw Data | SFT |
|---|---|---|---|---|---|---|
| FSHL(Liu et al., 2023) | Token | CL | ✔ | ✔ | ✔ | |
| Gruver et al. (2023) | Token | FC | ✔ | | ✔ | |
| HealthLLM (Kim et al., 2024) | Token | TR | ✔ | ✔ | ✔ | ✔ |
| Chow et al. (2024) | SP | TR | ✔ | ✔ | ✔ | ✔ |
| ChatTS (Xie et al., 2025) | SP | TR | ✔ | ✔ | ✔ | ✔ |
| ITFormer (Wang et al., 2025b) | SP | TR | ✔ | ✔ | ✔ | ✔ |
| InstrucTime (Cheng et al., 2025) | SP | TR | ✔ | ✔ | ✔ | ✔ |
| MedTsLLM (Chan et al., 2024) | SP | CL | | ✔ | ✔ | |
| MedualTime (Ye et al., 2025) | SP | CL | | ✔ | ✔ | |
| SensorLLM (Li et al., 2025) | SP | CL | | ✔ | ✔ | |
| Time2Lang (Pillai et al., 2025) | SP | CL | | ✔ | | |
| **OpenTSLM-SP (ours)** | SP | TR | ✔ | ✔ | ✔ | ✔ |
| SensorLM (Zhang et al., 2025) | XAttn | CL | ✔ | ✔ | | |
| **OpenTSLM-Flamingo (ours)** | XAttn | TR | ✔ | ✔ | ✔ | ✔ |

Token =Tokenization, SP =Soft Prompting, XAttn =Cross-Attention, Adapter =Adapter; CL =Classification, FC =Forecasting, TR =Text Reasoning

### 2.1. Tokenization of Time-Series as Text Inputs

Gruver et al. (2023) has demonstrated that LLMs can perform time-series forecasting by encoding values as text tokens and predicting future values without domain-specific tuning. Liu et al. (2023) tokenize data from wearables and smartphones to enable LLMs to infer clinical and wellness information through few-shot prompting. Similarly, Kim et al. (2024) propose HealthLLM, a framework for health prediction using physiological signals (e.g., heart rate, sleep) combined with user context and medical knowledge embedded in prompts.

### 2.2. Combining Text and Time-Series Token Embeddings (Soft Prompting)

An alternative to manual tokenization is to encode time-series into embeddings that capture time-series information, using a time-series encoder as presented by Nie et al. (2023). These embeddings can be input into a transformer directly or concatenated with text embeddings (soft prompting) (Chow et al., 2024; Nie et al., 2023; Pillai et al., 2025; Ye et al., 2025; Xie et al., 2025; Wang et al., 2025b; Cheng et al., 2025; Chan et al., 2024). Pillai et al. (2025) use this approach and train an encoder to produce soft prompts from time-series, which are then processed by a frozen LLM for classification via a projection head; however, this disables free-form text generation. Ye et al. (2025) and Chan et al. (2024) similarly combine time-series and text-token embeddings, using a classification head (Ye et al., 2025) and a task solver (Chan et al., 2024) for prediction. Wang et al. (2025b) introduced ITFormer, a novel framework that

combines any time-series encoder with any frozen LLM to support time-series question answering, also by combining text- and derived time-series tokens. Cheng et al. (2025) introduce a framework that first aligns time-series and natural language in a general stage, and later finetunes for a specific domain to perform classification. Li et al. (2025) integrate sensor and text embeddings in two stages: First, they generate a caption-like summary of the time-series for free-form output; Second, they classify the data via a projection head, therefore restricting free from output. Chow et al. (2024) and Xie et al. (2025) interleave time-series tokens with text tokens in the LLM input, enabling free-form text reasoning.

### 2.3. Cross-Attention for Time-Series Data

Few studies use cross-attention to integrate time-series into LLMs. Zhang et al. (2025) apply cross-attention between a time-series encoder and a text encoder, aligned with contrastive loss, to extract statistical summaries (e.g., mean, max) from a single sensor. They train a new sensor encoder, text encoder, and multimodal text decoder, rather than adapting a pretrained LLM (Zhang et al., 2025).

### 2.4. Pretrained Time-Series Foundation Models

Recent work on time-series foundation models (TSFMs) shows that large-scale pretraining enables strong zero-shot forecasting across domains. TimesFM (Das et al., 2024) proposes a practical decoder-only Transformer with patching, pretrained on large-scale real-world time-series data, and achieves competitive performance. Moirai2 (Liu et al., 2025) adopts a decoder-only design with quantile-based objectives and multi-token prediction to improve probabilistic accuracy and inference efficiency. Chronos-2 (Ansari et al., 2025) extends pretrained forecasting to multivariate and covariate-aware settings via group attention for sharing information across related series. While these TSFMs achieve strong results on established forecasting benchmarks (e.g., GIFT-Eval (Aksu et al., 2024)), they are primarily designed for forecasting and are typically not based on LLMs; hence, they lack explicit reasoning capabilities and text generation interfaces.

## 3. Methods

We present two architectures for TSLMs, OpenTSLM-Soft Prompting (SP) (Section 3.2) and OpenTSLM-Flamingo (Section 3.3). To support multiple time-series inputs, we design a prompt format that interleaves sensor data with accompanying textual descriptions (e.g., "Data from Sensor X over Y days:" followed by the data representation). Figure 1 illustrates our approach.

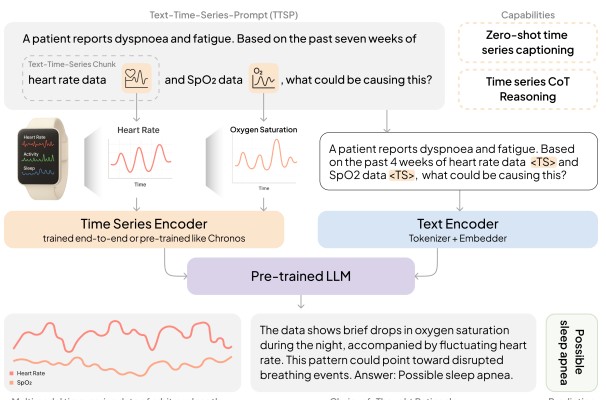

*Figure 1.* Overview of Text–Time-Series LLMs with support for multiple time-series inputs.

### 3.1. Time-Series encoder

We use a time-series encoder to extract features that characterize the input time-series. We support either training an encoder from scratch, leveraging the PatchTST architecture (Nie et al., 2023), or using pretrained TSFMs such as Chronos-2 (Ansari et al., 2025). When trained from scratch, the encoder is trained with our models, whereas for pretrained encoders the parameters are frozen and only a projection layer is trained alongside our models (see Section 4.1).

In the case of PatchTST, the encoder consists of a `PatchEncoder`, followed by either a `TransformerEncoder` for OpenTSLM-SP or a `PerceiverResampler` for OpenTSLM-Flamingo, inspired by Alayrac et al. (2022); Awadalla et al. (2023). We divide an input time-series $x \in \mathbb{R}^L$ into non-overlapping patches of size $p$, yielding $N = L/p$ patches. Each patch is then transformed into a patch embedding vector using a 1D convolution and combined with a positional encoding (Nie et al., 2023):

$$\mathbf{E}i = \text{Conv1D}(xi \cdot p : (i+1) \cdot p) \in \mathbb{R}^{d_{\text{enc}}} + \mathbf{P}i, \quad (1)$$

where the convolution has kernel size and stride equal to $p$, mapping each patch to a $d_{\text{enc}}$-dimensional embedding, and $\mathbf{P}_i$ denotes a learnable positional encoding. The resulting sequence of position-augmented embeddings is processed by the corresponding encoder (cf. Sections 3.2 and 3.3).

**Preserving scale and temporal information** The `PatchEncoder` expects inputs normalized to $x \in [-1, 1]$. Since raw time-series differ in scale and resolution across modalities depending on the sensor. Consistent with prior work (Chow et al., 2024; Xie et al., 2025) we preserve scale and temporal context by adding the original mean, standard deviation, and time scale to the textual description. For example: *This is heart-rate data over **24 hours sampled at 50 Hz** with **mean=61** and **std=12**.*

*Figure 2.* Architecture of OpenTSLM-SoftPrompt

## 3.2. Soft prompting architecture (OpenTSLM-SP)

OpenTSLM-SP has three components (Figure 2): (1) a time-series encoder that transforms raw data into patch embeddings, (2) a projection layer mapping embeddings to the LLM hidden space, (3) a pretrained LLM, fine-tuned using LoRA adapters (Hu et al., 2021).

**Projecting Time-Series Tokens to Text Tokens**  We apply the patch embeddings to a transformer encoder and subsequently project the resulting tokens with an multi-layer perceptron (MLP) to align them with the embedding space of dimension $d_{\text{llm}}$, corresponding to the hidden size of the LLM, following Nie et al. (2023) and Chow et al. (2024).

$$\mathbf{Z} = \text{MLP}(\mathbf{TransformerEncoder}(E_{1:N})) \in \mathbb{R}^{N \times d_{\text{llm}}} \tag{2}$$

where $\mathbf{Z} \in \mathbb{R}^{N \times d_{\text{llm}}}$ denotes the projected time-series tokens in the LLM embedding space.

**Text-Time-Series integration via Soft Prompting**  We interleave any number of text and time-series tokens through a soft prompting mechanism. A typical prompt consists of (1) an initial text segment ("pre-prompt"), (2) a sequence of interleaved time-series tokens and textual descriptions, and (3) a final text segment ("post-prompt"), often a question. Formally, the model input is:

$$\mathbf{X}_{\text{input}} = [\mathbf{T}_{\text{pre}}, \mathbf{Z}_1, \mathbf{T}_{\text{desc}_1}, \mathbf{Z}_2, \mathbf{T}_{\text{desc}_2}, \ldots, \mathbf{Z}_K, \mathbf{T}_{\text{desc}_K}, \mathbf{T}_{\text{post}}] \tag{3}$$

where $\mathbf{T}_{\text{pre}}$, $\mathbf{T}_{\text{desc}_i}$, and $\mathbf{T}_{\text{post}}$ are token embeddings of text segments, and each $\mathbf{Z}_i$ is a projected time-series embedding aligned with the LLM hidden space. We refer to each $(\mathbf{Z}_i, \mathbf{T}_{\text{desc}_i})$ as a text–time-series chunk. This approach implicitly integrates time-series through learned tokens.

## 3.3. Cross-attention architecture (OpenTSLM-Flamingo)

OpenTSLM-Flamingo is inspired by the Flamingo model for vision–language tasks (Alayrac et al., 2022; Awadalla et al., 2023). Following OpenFlamingo (Awadalla et al., 2023), we extend pretrained LLMs with cross-attention layers to support time-series reasoning.

**Architecture Overview**  We replace the vision encoder of Flamingo with a time-series encoder and adapt the cross-attention mechanism for temporal data. The model consists of: (1) a time-series patch encoder, (2) a Perceiver Resampler, (3) gated cross-attention layers integrated into the LLM, and (4) the frozen language model backbone. Figure 3 visualizes the architecture.

**PerceiverResampler**  We use a PerceiverResampler inspired by Flamingo (Awadalla et al., 2023) as Encoder for the time-series patches, yielding a fixed-size latent representation:

$$\mathbf{Z}_{\text{latent}} = \text{PerceiverResampler}(\mathbf{E}_{1:N}) \in \mathbb{R}^{N_{\text{latent}} \times d_{\text{time}}}, \tag{4}$$

Here, $d_{\text{time}}$ is the dimensionality of the time-series features by the perceiver, in our case $(N, 1)$, encoding one time-series with one channel at a time.

**Text-Time-Series Gated Cross-Attention**  To integrate $\mathbf{Z}_{\text{latent}}$ into the LLM, we add gated cross-attention layers every $N$ (hyperparameter) transformer blocks which compute:

$$\mathbf{Q}_{\text{text}} = \mathbf{x}\mathbf{W}_Q, \quad \mathbf{K}_{\text{ts}} = \mathbf{Z}_{\text{latent}}\mathbf{W}_K, \quad \mathbf{V}_{\text{ts}} = \mathbf{Z}_{\text{latent}}\mathbf{W}_V \tag{5}$$

$$\text{Gated-XAttn}(\mathbf{x}, \mathbf{Z}_{\text{latent}}) = x + \gamma \cdot \text{softmax}\left(\frac{\mathbf{Q}_{\text{text}}\mathbf{K}_{\text{ts}}^T}{\sqrt{d_k}}\right)\mathbf{V}_{\text{ts}}. \tag{6}$$

where $\gamma_{\text{attn}}$ is a learnable parameter controlling the influence of the time-series, $\mathbf{x} \in \mathbb{R}^{T \times d_{\text{model}}}$, the LLM input, $\mathbf{W}_Q, \mathbf{W}_K, \mathbf{W}_V \in \mathbb{R}^{d_{\text{model}} \times d_k}$ learned projection matrices, and $d_k$ the key dimension.

**Conditioning Text-tokens on Time-Series via Special Tokens**  The LLM processes tokens autoregressively, attending to previous inputs. Following OpenFlamingo (Awadalla et al., 2023), we introduce special tokens $\langle\text{TS}\rangle$ and $\langle\text{endofchunk}\rangle$ to indicate when time-series modalities should be incorporated. Upon encountering $\langle\text{TS}\rangle$, the model conditions on the corresponding latent representation $\mathbf{Z}_{\text{latent}}$ via gated cross-attention. A typical input prompt is

$$\mathbf{X}_{\text{input}} = [\text{pre\_prompt}, \langle\text{TS}\rangle, \text{ts\_desc}_1, \langle\text{endofchunk}\rangle, \\ \langle\text{TS}\rangle, \text{ts\_desc}_2, \langle\text{endofchunk}\rangle, \text{post\_prompt}] \tag{7}$$

*Figure 3.* Architecture of OpenTSLM-Flamingo

where ⟨TS⟩ triggers multimodal conditioning and ⟨endofchunk⟩ signals the end of text describing a time-series. This setup enables interleaving multiple text and time-series segments (Awadalla et al., 2023). The embeddings of the special tokens are learned.

## 4. Experiments

In the following, we outline our training methodology and report results on multiple-choice Time-Series Question Answering (TSQA) and time-series reasoning datasets. We compare OpenTSLM-SoftPrompt and OpenTSLM-Flamingo against each other and baselines in terms of performance, and report video random access memory (VRAM) requirements for training OpenTSLM. We present sample model outputs across datasets and an evaluation for electrocardiogram (ECG) rationales by expert cardiologists.

### 4.1. Multi-Stage Curriculum Learning – Teaching LLMs Time-Series

Following (Chow et al., 2024), we adopt a two-stage curriculum to train TSLMs. In stage one (encoder warmup), we use two synthetic time-series datasets to pretrain the encoder:

- **TSQA (Wang et al., 2024)** Multiple-choice time-series question answering on synthetic data for learning simple temporal patterns (e.g., ascending/descending trends).
- **Time-Series Captioning (M4-Captions)** We generate pseudo-labeled captions using ChatGPT, prompted with plots of time-series of the M4 dataset (Makridakis et al., 2020) (see Section A.5.1).

In stage two, we introduce three new CoT time-series datasets covering human activity recognition (HAR), sleep staging, and ECG Question Answering (QA). We generated these using GPT-4o by providing a plot and ground-truth answer for each sample, then asking the model to produce rationales leading to the correct response. Further details are provided in Section A.2.

- **HAR-CoT** three-axis accelerometer data combined from DaLiAc (Leutheuser et al., 2013), DOMINO (Arrotta et al., 2023), HHAR (Stisen et al., 2015), PAMAP2 (Reiss & Stricker, 2012), RealWorld (Sztyler

& Stuckenschmidt, 2016), and datasets from (Shoaib et al., 2013; 2014; 2016). Sampled at 50 Hz, split into 2.56s windows, 8 activities: sitting, standing, lying, walking, running, biking, walking upstairs, walking downstairs. See Section A.2.1 for detailed description.

- **Sleep-CoT** Based on SleepEDF (Kemp et al., 2000; Goldberger et al., 2000), using 30s electroencephalogram (EEG) segments for sleep staging. Following prior work (Chow et al., 2024; Pouliou et al., 2025), Non-rapid eye movement (REM) stages 3 and 4 are merged, yielding five classes: Wake, REM, Non-REM1, Non-REM2, Non-REM3. See Section A.2.2 for details.
- **ECG-QA-CoT** Based on ECG-QA (Oh et al., 2023), which provides 12-lead 10s ECGs and clinical context, we excluded comparison questions, retaining 42/70 templates. This yielded 3,138 unique questions across 240k samples (see Section A.2.3).

All datasets are split into **80/10/10 train/validation/test** sets. Table 7 in Section A.1 summarizes number of samples in the datasets, number of time-series and lengths.

**Training objective** In all stages, we frame the task as an autoregressive language modeling problem. During training and evaluation, the model is prompted to generate outputs in a structured format, consisting of a free-form rationale followed by the final prediction: `<reasoning> Answer: <final answer>`. Formally, the loss is defined by Equation 8, where $\mathbf{Z}_{\text{ts}}$ are the

$$\mathcal{L}_{\text{LM}} = -\sum_{t=1}^{T} \log P(y_t \mid y_{<t}, \mathbf{x}_{1:t}, \mathbf{Z}_{\text{ts}}; \Theta) \qquad (8)$$

time-series features, and $\Theta$ the learnable weights, i.e., the TimeSeriesEncoder, MLP, and LoRA in OpenTSLM-SoftPrompt, and TimeSeriesEncoder and cross-attention in OpenTSLM-Flamingo.

### 4.2. Baselines

We compare OpenTSLM against three baselines using the same open-weight LLMs, i.e., Llama-3.2(1B, 3B) and Gemma3 (270M, 1B-PT), and additionally GPT-4o (gpt-4o-2024-08-06).

1. **Tokenized time-series**: Using the open-source code provided by Gruver et al. (2023), we tokenize time-series into text inputs and report zero-shot performance on the test set.
2. **Tokenized finetuned**: Same as 1. (excluding GPT-4o), but finetuned with LoRA (Hu et al., 2021) on the training set. We choose best model by validation loss, and report performance on test set.
3. **Image (Plot)**: We convert time-series into as input for GPT-4o and Gemma-4b-pt (since the smaller Gemma 3 variants do not support image input).
4. **Random baseline**: For comparison, we report the expected performance of a predictor that selects labels uniformly at random, adjusted to each dataset's label distribution.

### 4.3. Quantitative results on time-series classification

We present performance on the test splits of TSQA, HAR-CoT, Sleep-CoT, and ECG-QA-CoT and report macro-F1 score and accuracy in Table 2. OpenTSLM models per-

*Table 2.* Performance comparison on time-series question answering (TSQA) and time-series reasoning (HAR-CoT, Sleep-CoT, ECG-QA-CoT) tasks between OpenTSLM models and baselines.

| | Model | TSQA F1 | TSQA Acc | HAR-CoT F1 | HAR-CoT Acc | Sleep-CoT F1 | Sleep-CoT Acc | ECG-QA-CoT F1 | ECG-QA-CoT Acc |
|---|---|---|---|---|---|---|---|---|---|
| | Random Baseline | 33.33 | 33.33 | 11.49 | 12.50 | 17.48 | 20.00 | 16.47 | 20.18 |
| Tokenized Time-Series | Llama3.2-1B | 16.01 | 31.04 | 0.00[*1] | 0.00 | 2.14 | 0.65 | 0.00 | 0.00 |
| | Llama3.2-3B | 16.24 | 32.06 | 0.00 | 0.00 | 5.66 | 12.15 | 0.00 | 0.00 |
| | Gemma3-270M | 10.52 | 9.58 | 0.00 | 0.00 | 0.00 | 0.00 | 0.00 | 0.00 |
| | Gemma3-1B-pt | 11.76 | 12.92 | 0.00 | 0.00 | 0.00 | 0.00 | 0.00 | 0.00 |
| | GPT-4o | 45.32 | 45.29 | 2.95 | 11.74 | 15.47 | 16.02 | 18.19 | 28.76 |
| Tokenized Finetuned | Llama3.2-1B | 83.74 | 81.40 | 51.28 | 62.71 | 9.05 | 24.19 | OOM | OOM[*2] |
| | Llama3.2-3B | 84.54 | 82.06 | 60.44 | 66.87 | 5.86 | 14.30 | OOM | OOM |
| | Gemma3-270M | 68.05 | 65.40 | 40.66 | 54.56 | 0.00 | 0.00 | OOM | OOM |
| | Gemma3-1B-pt | 82.85 | 83.42 | 52.15 | 63.90 | 0.00 | 0.00 | OOM | OOM |
| Image (Plot) | Gemma3-4B-pt | 48.77 | 50.60 | 1.72 | 0.89 | 6.75 | 14.95 | 1.90 | 1.03 |
| | Gemma3-4B FT | 94.94 | 94.85 | 44.01 | 51.02 | 18.56 | 38.28 | 26.17 | 38.10 |
| | GPT-4o | 59.24 | 62.10 | 10.83 | 13.90 | 4.82 | 10.75 | 24.95 | 33.30 |
| OpenTSLM SoftPrompt | Llama3.2-1B | **97.50** | **97.54** | **65.44** | **71.48** | **69.88** | **81.08** | 32.84 | 35.49 |
| | Llama3.2-3B | 97.37 | 97.33 | 64.87 | 67.89 | 54.40 | 72.04 | 33.67 | 36.25 |
| | Gemma3-270M | 40.32 | 26.79 | 1.43 | 0.55 | 7.96 | 5.91 | 1.29 | 1.11 |
| | Gemma3-1B-pt | 87.29 | 89.18 | 40.52 | 45.17 | 30.99 | 36.56 | 27.86 | 34.76 |
| OpenTSLM Flamingo | Llama3.2-1B | 94.08 | 94.00 | 62.93 | 69.27 | 49.33 | 67.31 | 34.62 | 38.14 |
| | Llama3.2-3B | 90.14 | 90.10 | 62.77 | 69.03 | 45.45 | 69.14 | **40.25** | **46.25** |
| | Gemma3-270M | 77.86 | 78.12 | 57.75 | 63.43 | 51.38 | 68.49 | 32.71 | 35.50 |
| | Gemma3-1B-pt | 92.56 | 92.46 | **65.44** | **71.48** | 43.69 | 60.67 | 35.31 | 37.79 |

Note: Gemma models have smaller context than Llama (32k vs. 128k); soft prompt uses up context, performing worse. [*1]0.00 model failed to produce Answer: {answer} template, often repeating input prompt (see Section A.4).[*2]OOM - Out of memory: 12 ECG leads of 10s tokenize to 80k tokens, requiring >100GB VRAM. Gemma3-4B FT refers to Gemma-3-4B-pt finetuned.

form best across benchmarks, while most tokenized text-only baselines fail to produce valid outputs, not answering in the expected template but merely repeating inputs or

starting to count (see Section A.4), resulting in 0.00 F1 on HAR for all models except for GPT-4o (2.95). GPT-4o improves substantially with plots (e.g., 10.83 on HAR, 59.24 on TSQA). Using plot inputs, Gemma3-4B achieves 48.77 F1 on TSQA, 1.72 on HAR-CoT, 6.75 on Sleep-CoT, and 1.90 on ECG-QA-CoT, but improves significantly after fine-tuning, achieving 94.94,0 44.01, 18.56, and 26.17, respectively. Llama models achieve 2.14 and 5.65F1 on Sleep, while Gemma models remain at 0.00, likely due to their smaller context window (32k vs. 128k). By contrast, OpenTSLM–SoftPrompt with Llama3.2-1B attains 97.50 F1 (97.54 accuracy) on TSQA, with Llama3.2-3B at 97.37 (97.33); Flamingo variants are close (e.g., Llama3.2-1B 94.08 (94.00)). The strongest tokenized-finetuned baseline reaches 84.54 (82.06) and GPT-4o with plots achieves 59.24 (62.10). On HAR-CoT, the strongest results are 65.44F1 (71.48 accuracy) for OpenTSLM–SoftPrompt (Llama3.2-1B) and 65.44 (71.48) for OpenTSLM–Flamingo (Gemma3-1B); the best tokenized-finetuned baseline achieves 60.44 (66.87). On Sleep-CoT, OpenTSLM–SoftPrompt (Llama3.2-1B) achieves 69.88 (81.08), followed by OpenTSLM–SoftPrompt (Llama3.2-3B) at 54.40 (72.04) and Flamingo (Gemma3-270M) at 51.38 (68.49); tokenized-finetuned baselines remain lower at max. 9.05 (24.19). On ECG-QA-CoT, OpenTSLM–Flamingo (Llama3.2-3B) leads with 40.25 (46.25).

### 4.4. Evaluation of memory use during training

We evaluate peak VRAM usage during training for both OpenTSLM variants. Figure 4 summarizes peak VRAM on TSQA, HAR–CoT, SleepEDF–CoT, and ECG–QA–CoT. OpenTSLM-Flamingo shows near-constant memory across

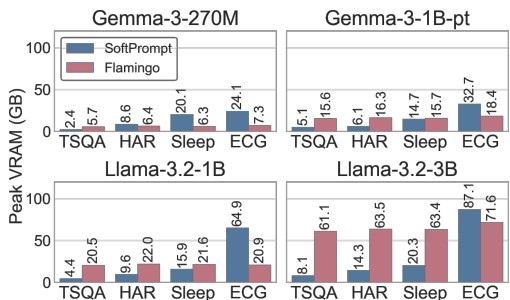

*Figure 4.* VRAM memory usage in training across datasets.

datasets, while OpenTSLM-SoftPrompt vary substantially with the dataset, suggesting that the memory cost of OpenTSLM-SoftPrompt is highly related to the length of time-series and the text tokens.

To further investigate memory scaling, we train models on a simulated dataset (see Section A.7.2) with various input shapes. We report max VRAM usage in the appendix (see Figure 18 and Table 14). We observe that VRAM usage of

OpenTSLM-Flamingo stays constant as the number of time-series $N$ and the sequence length $L$ increase. In contrast, the VRAM usage of OpenTSLM-SoftPrompt scales with both $N$ and $L$, suggesting poor memory scalability, which can become a practical bottleneck for longer contexts.

### 4.5. Qualitative results and expert evaluation of ECG rationales

Both **OpenTSLM** variants remain text models, trained to generate rationales for classification rather than outputting only a class label. Figure 5a shows example rationales for *human activity recognition* and Figure 5b for ECG-QA. Figure 5c shows time-series captioning on M4 data.

To evaluate the quality of model rationales, we conducted an expert review with five cardiologists from Stanford Hospital on rationales generated by OpenTSLM-Flamingo-Llama3.2-3B (best model) for ECG-QA. We randomly sampled two examples per template (84 total), each reviewed by at least two cardiologists. Evaluation followed a rubric derived from the American College of Cardiology/American Heart Association Clinical Competence Statement on ECGs (Pangaro, 1999; Members et al., 2001) and based on the RIME ("Reporter–Interpreter–Manager–Educator") framework (Pangaro, 1999) (see A.6), assessing whether the model: (1) correctly identified relevant ECG features; (2) appropriately connected them to the final answer; (3) incorporated patient context (age, artifacts, ...). Overall, the model gave a correct or partially correct ECG interpretation in 92.9% of cases, spanning ECG recognition, reasoning, and contextualization. OpenTSLM showed strongest performance in clinical context integration (85.1% positive) compared to ECG pattern recognition (65.5% positive) and clinical reasoning (62.5% positive) (Figure 6a). Assessment patterns varied notably across reviewers, with some reviewers consistently more favorable across all evaluation areas (Figure 6b). Reviewer disagreement was most common for clinical reasoning, where moderate disagreements were observed between adjacent assessment categories. Complete disagreements between positive and negative assessments were relatively rare across all areas (Figure 16 in A.6).

### 4.6. OpenTSLM with pretrained time-series encoder

Motivated by the strong performance of existing pretrained TSFMs, such as Chronos-2 (Ansari et al., 2025), we further investigate their compatibility with the OpenTSLM framework (see Section 3.1). Table 3 shows that using OpenTSLM-Flamingo with pretrained Chronos-2 generally improves performance across all LLM backbones and datasets. In particular, for TSQA, even relatively small LLMs (e.g., Gemma3-270M) benefit substantially, with performance improving from 77.86% to 99.69%. For Sleep-CoT, pretrained TSFMs lead to substantial F1 gains across

all backbones, e.g., improvements from 49.33 to 64.57 for Llama3.2-1B and from 51.38 to 68.33 for Gemma3-270M. We also observe moderate F1 improvements on ECG-QA-CoT (e.g., from 34.62 to 41.92 for Llama3.2-1B).

*Table 3.* OpenTSLM-Flamingo with different LLM bases and a pretrained encoder time-series encoder.

| | TSQA | | HAR-CoT | | Sleep-CoT | | ECG-QA-CoT | |
|---|---|---|---|---|---|---|---|---|
| | F1 | Acc | F1 | Acc | F1 | Acc | F1 | Acc |
| Llama3.2-1B | 94.08 | 94.00 | 62.93 | 69.27 | 49.33 | 67.31 | 34.62 | 38.14 |
| Llama3.2-1B-Chronos2 | **99.71** | **99.71** | 66.82 | 70.61 | 64.57 | 78.54 | **41.92** | 44.01 |
| Llama3.2-3B | 90.14 | 90.10 | 62.77 | 69.03 | 45.45 | 69.14 | 40.25 | **46.25** |
| Llama3.2-3B-Chronos2 | 99.58 | 99.58 | **67.64** | **72.20** | 61.52 | 78.31 | 41.89 | 44.27 |
| Gemma3-270M | 77.86 | 78.12 | 57.75 | 63.43 | 51.38 | 68.49 | 32.71 | 35.50 |
| Gemma3-270M-Chronos2 | 99.69 | 99.69 | 58.65 | 63.90 | **68.33** | **80.69** | 35.92 | 38.37 |
| Gemma3-1B-pt | 92.56 | 92.46 | 65.44 | 71.48 | 43.69 | 60.67 | 35.31 | 37.79 |
| Gemma3-1B-pt-Chronos2 | **99.71** | **99.71** | 62.72 | 67.50 | 61.13 | 75.96 | 40.25 | 42.47 |

### 4.7. Cross-task generalization

To investigate whether training across heterogeneous time-series tasks improves generalization and downstream performance on individual tasks, we train OpenTSLM-Flamingo on the merged collection of all training datasets and compare it against the curriculum-learning setup. We use OpenTSLM with Llama3.2-3B, assuming that more capacity is required to learn shared representations across tasks. Despite differ-

*Table 4.* Cross-task generalization performance (F1) for OpenTSLM-Flamingo with Llama3.2-3B.

| | TSQA | HAR-CoT | Sleep-CoT | ECG-QA-CoT |
|---|---|---|---|---|
| Llama3.2-3B single-task | 90.14 | 62.77 | 45.45 | 40.25 |
| Llama3.2-3B merged | **92.73** | **69.31** | **52.13** | **41.55** |
| Improvement | +2.59 | +6.54 | +6.68 | +1.30 |

ences in time-series domains, sequence lengths, and sensor configurations, joint training improves performance across all datasets compared to curriculum learning, as shown in Table 4. The largest gains are observed on HAR-CoT and Sleep-CoT, with improvements of 6.54 and 6.68 in F1, respectively. We also observe smaller but consistent improvements on TSQA and ECG-QA-CoT.

### 4.8. Noise ablation

To assess robustness in situations with noisy or corrupted input, we evaluate OpenTSLM-Flamingo under two controlled perturbation settings. First, we add Gaussian noise to the input time-series by linearly interpolating the original signal $\mathbf{x}$ with noise $\epsilon \sim \mathcal{N}(0, 1)$:

$$\tilde{\mathbf{x}} = (1 - \alpha)\mathbf{x} + \alpha\epsilon, \qquad (9)$$

where $\alpha$ controls the noise level. Second, we simulate sensor failure by replacing the full input with Gaussian noise.

As shown in Table 5, performance generally degrades as input noise increases, but the model does not collapse and

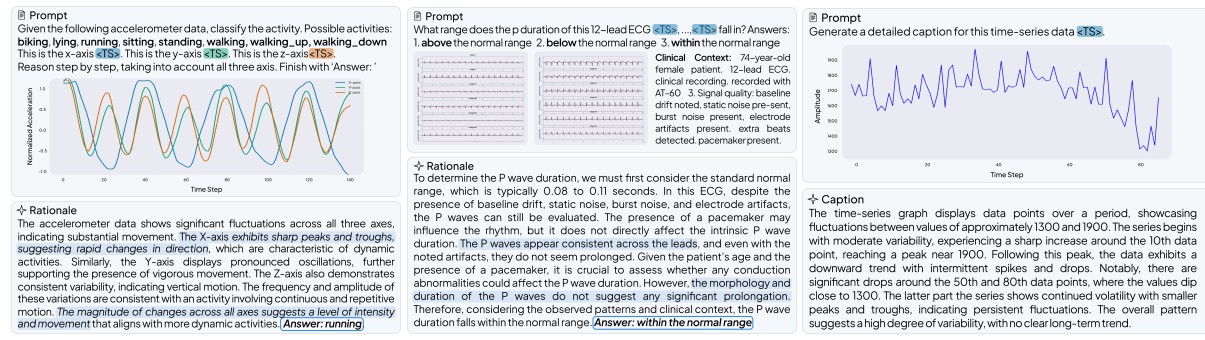

*(a)* Human Activity Recognition CoT   *(b)* ECG Interpretation CoT   *(c)* M4 Time-Series Captioning

*Figure 5.* Example CoT rationales for HAR, Sleep Staging, ECG-QA and M4 captioning, generated with OpenTSLM-Flamingo/Llama3.2-1B. More examples are provided in Section A.5.



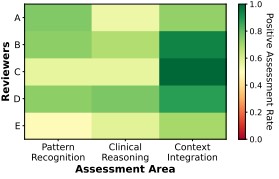

*(a)* Performance by Area   *(b)* Eval Distribution

*Figure 6.* Qualitative evaluation of CoT rationales and inter-reviewer agreement patterns.

*Table 5.* Noise ablation F1 scores for OpenTSLM-Flamingo

| Noise level | TSQA | HAR-CoT | Sleep-CoT | ECG-QA-CoT |
|---|---|---|---|---|
| Normal | 94.08 | 62.93 | 49.33 | 41.58 |
| 10% | 91.55 | 62.91 | 43.74 | 40.27 |
| 30% | 77.11 | 55.56 | 31.27 | 39.80 |
| 50% | 50.82 | 43.10 | 12.76 | 39.92 |
| 70% | 35.28 | 34.15 | 12.46 | 39.46 |
| Sensor failure | 32.48 | 10.59 | 16.50 | 28.31 |

continues to produce valid outputs. We observe the strongest degradation for TSQA, HAR-CoT, and Sleep-CoT, while ECG-QA-CoT remains comparatively stable under additive noise, which might indicate that the model uses the additionally provided patient context to ground its responses. Under complete sensor failure, performance drops more substantially, suggesting that predictions may increasingly reflect learned priors rather than grounded time-series reasoning.

### 4.9. Performance comparison with related models

We compare OpenTSLM against ITFormer (Wang et al., 2025b), InstructTime (Cheng et al., 2025), ChatTS (Xie et al., 2025), and ChatTime (Wang et al., 2025a) on the three datasets across all model variants. As shown in Table 6, OpenTSLM outperforms all related models across the three datasets. OpenTSLM-SoftPrompt with Llama3.2-1B achieves the best results on HAR-CoT and Sleep-CoT, whereas OpenTSLM-Flamingo with Llama3.2-3B achieves

*Table 6.* Comparison with related models (F1/Accuracy)

| Model | HAR-CoT | | Sleep-CoT | | ECG-QA-CoT | |
|---|---|---|---|---|---|---|
| | F1 | Acc. | F1 | Acc. | F1 | Acc. |
| ITFormer | 55.41 | 58.06 | 42.38 | 70.21 | 37.52 | 39.98 |
| InstructTime | 61.73 | 65.07 | 42.57 | 59.31 | 20.71 | 25.16 |
| ChatTS | 57.63 | 64.05 | 41.31 | 73.23 | 8.91 | 42.32 |
| ChatTime | 41.11 | 63.28 | 19.09 | 43.12 | OOM | OOM |
| *OpenTSLM SoftPrompt* | | | | | | |
| Llama3.2-1B | **65.44** | **71.48** | **69.88** | **81.08** | 32.84 | 35.49 |
| Llama3.2-3B | 64.87 | 67.89 | 54.40 | 72.04 | 33.67 | 36.25 |
| *OpenTSLM Flamingo* | | | | | | |
| Llama3.2-1B | 62.93 | 69.27 | 49.33 | 67.31 | 34.62 | 38.14 |
| Llama3.2-3B | 62.77 | 69.03 | 45.45 | 69.14 | **40.25** | **46.25** |

the best result on ECG-QA-CoT.

## 5. Discussion

All OpenTSLM models consistently outperform baselines. Text-only models often fail to follow the answer template and thus perform at or below chance (Section 4.1). Fine-tuned baselines improve substantially on HAR-CoT (60.44% F1 vs. 0% for Llama-3.2-1B) but only slightly on Sleep-CoT (9.05 vs. 2.14). ECG-QA finetuning was infeasible due to high VRAM demands (80k tokens require >100GB per sample). OpenTSLM-SoftPrompt performs best on shorter sequences (Sleep-CoT, TSQA) but becomes impractical as VRAM requirements grow with sequence length (>180GB in simulations with 10,000-length series). With soft prompting, smaller models like Gemma-3 270M and 1B quickly exhaust their context and underperform. In contrast, OpenTSLM-Flamingo sustains stable memory across sequence lengths and series (up to 60GB for Llama-3.2-3B with five 10,000-length series). This allows even tiny models, such as Gemma-270M, to deliver strong results, highlighting the efficiency of cross-attention for treating time-series as a native modality. We also find that joint training across heterogeneous time-series tasks can improve performance and generalization on individual downstream

tasks, with the jointly trained OpenTSLM-Flamingo Llama 3B model increasing increasing performance between 1.30% and 6.68% depending on the dataset (see Section 4.7). These results suggest potential for further scaling and improved generalization by training larger models on more diverse time-series data. Leveraging pretrained TSFMs such as Chronos-2 further improves performance, in some cases substantially, for example, TSQA accuracy increases from 77.86% to 99.69% with Gemma-3 270M. OpenTSLM is compatible with TSFMs via a trainable projection layer.

**Practical implications.** Our results show that even frontier LLMs like GPT-4o are poorly suited for time-series reasoning and that time-series must be treated as a distinct modality. With OpenTSLM, even small models like Gemma3 270M outperform GPT-4o (∼200B parameters (Abacha et al., 2025)) at a fraction of the compute and cost, enabling efficient on-device or mobile deployment. We recommend using OpenTSLM-SoftPrompt for short time-series, where it shows strong performance while requiring only a small number of additional parameters for finetuning. However, because SoftPrompt's memory usage grows exponentially with sequence length, it becomes impractical for longer horizons or multi-series inputs. In contrast, we recommend OpenTSLM-Flamingo for longer time-series and multivariate sensor data (e.g, 12-lead ECG) and as a general-purpose solution, as it maintains nearly constant memory consumption across extended or multi-series contexts, and offers better performance on complex datasets (like ECG-QA). Perhaps the greatest advantage of TSLMs is the interface they provide for contextualizing results. In ECG-QA, OpenTSLM correctly identified the relevant ECG features in most cases, with missing context only 7.1% of the time. The model demonstrated particularly strong clinical context integration (85.1% positive assessments), thereby offering clinicians and researchers a transparent window into the model's reasoning. This transparency underscores the value of applying LLMs to time-series for medicine.

**Comparison with prior work.** Our approach differs from prior work in several ways. First, we introduce time-series as a native modality for LLMs, unlike (Sivarajkumar & Wang, 2023) and (Kim et al., 2024), which tokenize time-series. Second, we frame tasks as joint text–time-series reasoning, training models to generate rationales that integrate temporal information. This contrasts with MedualTime (Ye et al., 2025) and Time2Lang (Pillai et al., 2025), which reprogrammed LLMs with task-specific classification or forecasting heads, removing language generation capabilities. Notably, OpenTSLM achieves 40.25 F1 on ECG-QA-CoT, producing rationales across 3,138 questions and 42 templates with diverse answer options. By comparison, Ye et al. report 76 F1 on PTB-XL (underlying dataset of ECG-QA) with only four classes and a fixed classification head (Ye et al., 2025).

Third, unlike SensorLM (Zhang et al., 2025), which is trained from scratch, our models build on pretrained open-weight LLMs, retaining pretrained knowledge. Fourth, while prior work used soft prompting (Chow et al., 2024; Wang et al., 2025b) to model time-series implicitly by concatenating text-tokens with derived time-series tokens, we find that this approach scales poorly in memory use. In contrast, our OpenTSLM-Flamingo approach models time-series explicitly via a separate encoding integrated via cross-attention, scaling better to long sequences. However, this approach introduces a representation bottleneck, compressing time series to a fixed embedding which may need improvement for scaling to very long time series. Our quantitative comparison in Table 6 shows that OpenTSLM achieves the highest F1 across all three CoT datasets, outperforming related models like ITFormer, InstructTime, ChatTS and ChatTime.

**Limitations.** We acknowledge several limitations. First, our method of encoding time-series may not be optimal, as we rely on including the mean and standard deviation in accompanying texts to preserve the temporal scale. Second, we generated CoT datasets using GPT-4o on plots, which we have shown to perform poorly on these plots alone. Additionally, LLM-generated rationales may potentially entangle reasoning with mere explanation style-writing, which may require human oversight to verify. Third, framing tasks as natural language generation does not ensure that the model prioritizes the correct label after presenting its rationale, underscoring the need for loss functions that explicitly enforce correct answers. Fourth, we did not evaluate generalization to out of domain datasets. OpenTSLM models may not yet generalize to unseen domains or out of distribution datasets, which is an important direction for future work. Finally, we only assessed Chronos-2 as a pretrained TSFM and did not evaluate using other state-of-the-art models as encoders on performance, which we leave for future work.

## 6. Conclusion

Our results show that both OpenTSLM variants enable small-scale LLMs to outperform much larger text-only models on time-series tasks, demonstrating that lightweight, domain-adapted architectures can achieve strong performance without massive model scales. With OpenTSLM, we extend open-weight pretrained LLMs to process time-series, retaining knowledge while adapting them to temporal domains. This work may lay the foundation for general-purpose TSLMs capable of handling diverse time-series datasets. Although our focus is healthcare, the ability to reason over longitudinal data has broad relevance in domains such as finance, supply chain management, and industrial monitoring.

# Acknowledgements

We would like to thank Nicolas Zumarraga for his assistance in addressing reviewer comments during the rebuttal period, and Nikita Zubairov for his tremendous work in curating the HAR datasets through the integration of multiple data sources.

# Impact statement

**Applications.** OpenTSLM may enable a new class of applications in which users and AI agents can ask natural-language questions about raw multivariate time-series data in healthcare, including medical signals, vital signs and wearable stream. Beyond healthcare, similar models may enable new applications across industrial telemetry, robotics logs, financial signals, supply-chain data, and enterprise operations data. Such systems could support operational decision support, anomaly investigation, monitoring, forecasting explanations, and eventually more autonomous workflows that reason over temporal data.

**Social and economic impact.** By making complex temporal systems accessible through language, time-series language models could help clinicians, operators, researchers, and managers interpret patterns that would otherwise require specialized dashboards, manual inspection, or domain-specific models. This could improve reliability, efficiency, and responsiveness in healthcare, infrastructure, industrial systems, and enterprise operations.

**Ethical and safety considerations.** Because these models may inform decisions in high-stakes settings, incorrect interpretations or overconfident rationales could lead to harmful actions, especially under distribution shift, sensor failure, or noisy inputs. While the paper provides ablations in terms of robustness to noise in a healthcare setting, stronger guardrails would need to be implemented in production. Deployment should include uncertainty communication, human oversight, external validation, and safeguards against fully automated decisions based only on model-generated explanations.

**Misuse and governance.** The same capabilities could enable intrusive inference from physiological, behavioral, workplace, or infrastructure data, particularly when monitoring is performed without consent. Responsible use should require privacy protections, access controls, auditing, transparency about data collection and model limitations, and clear limits on autonomous decision-making uses.

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

# A. Appendix

## A.1. Training details

Table 7 provides an overview of the datasets used during training. All data was split into ratios of 80/10/10 (train/val/test).

| | Dataset | #Samples (Train/Val/Test) | Num series | Length | Frequency |
|---|---|---|---|---|---|
| **Stage 1** | TSQA[*1] | 38,400 / 4,800 / 4,800 | 1 | Hours to Years | Not specified |
| | M4-Captions | 80,000 / 10,000 / 10,000 | 1 | 64-512 points | Not specified |
| **Stage 2** | HAR-CoT | 68,542 / 8,718 / 8,222 | 3 | 2.56s | 50Hz |
| | Sleep-CoT | 7,434 / 930 / 930 | 1 | 30s | 100Hz |
| | ECG-QA-CoT | 159,313 / 31,137 / 41,093 | 12 | 10s | 100Hz |

*Table 7.* [*1]TSQA (Wang et al., 2024) Overview of datasets used in Stage 1 (pretraining tasks) and Stage 2 (task-specific CoT reasoning). Datasets are split in 80/10/10 ration.

### A.1.1. TRAINING CONFIGURATION

The models were trained with the following configuration:

- **Optimizer:** AdamW

- **Learning Rates:**
    - **OpenTSLM-SP:**
        * Time-series encoder: $2 \times 10^{-4}$
        * LoRA: $2 \times 10^{-4}$
        * Projector: $1 \times 10^{-4}$
    - **OpenTSLM-Flamingo:**
        * Encoder: $2 \times 10^{-4}$
        * Cross-attention layers: $2 \times 10^{-4}$

- **Scheduler:** Linear learning rate schedule with warmup

- **Warmup:** 10% of total training steps

- **Gradient Clipping:** $\ell_2$-norm capped at 1.0

- **Weight Decay:** 0.01

- **Training Length:** Up to 200 epochs with early stopping (patience = 5 epochs)

Learning rate choices were informed by (Chow et al., 2024). For pretrained Chronos-2 (Ansari et al., 2025), we directly download the model from https://huggingface.co/amazon/chronos-2. For learning rates for OpenTSLM-Flamingo with chronos-2, we keep as training from scratch.

## A.2. Generation of multivariate time-series CoT datasets

This section provides detailed descriptions of the CoT datasets generated for our study: Human Activity Recognition (HAR-CoT), Sleep Stage Classification (SleepEDF-CoT), and Electrocardiogram Question Answering (ECG-QA-CoT).

Our objective was to enable TSLMs not only to classify time-series but also to generate explicit reasoning chains. Since few datasets include CoT text, we generated our own multivariate time-series CoT datasets using widely adopted benchmarks in HAR, sleep staging, and ECG-QA, following a similiar approach as proposed by (Chow et al., 2024).

For each dataset, we generated rationales with GPT-4o by providing a plot of the data along with the correct label, and prompting the model to produce a rationale leading to that label. The exact prompts are described in Sections A.2.1,

A.2.2, and A.2.3. We carefully engineered the prompts and manually reviewed a subset of samples to ensure the generated rationales were consistent and sensible. When plotting, original data was used without normalization. If multiple time-series were present in a sample (e.g., three in HAR or twelve in ECG), all were plotted as separate subplots but combined into a single figure.

- **GPT-4o snapshot:** gpt-4o-2024-08-06

- **Temperature:** 0.3

- **Seed:** 42

The following subsections describe dataset-specific methodologies, data processing, prompts, answer selection, and final class distributions.

### A.2.1. HUMAN ACTIVITY RECOGNITON (HAR) CoT

We merged multiple HAR datasets spanning DaLiAc (Leutheuser et al., 2013), DOMINO (Arrotta et al., 2023), HHAR (Stisen et al., 2015), PAMAP2 (Reiss & Stricker, 2012), RealWorld (Sztyler & Stuckenschmidt, 2016), and datastes from (Shoaib et al., 2013; 2014; 2016). We retain only those activity classes present in all datasets. The final dataset includes eight activity classes: sitting, walking, standing, running, walking up stairs, walking down stairs, lying, and biking. The data is split into 2.56 second windows.

**Data Processing**  The dataset was processed to create 2.56-second windows of triaxial accelerometer data (X, Y, Z axes). Each sample was visualized as a multi-panel plot showing the acceleration signals across all three axes over the time window.

**Prompt for CoT generation**  We generated CoT rationales by prompting the model with a correct and dissimilar label. The following prompt template was used for HAR-CoT generation:

```
You are shown a time-series plot of accelerometer over a 2.56 second window.
This data corresponds to one of two possible activities:
[CORRECT_ACTIVITY]
[DISSIMILAR_ACTIVITY]

Your task is to classify the activity based on analysis of the data.

Instructions:
- Begin by analyzing the time-series without assuming a specific label.
- Think step-by-step about what the observed patterns suggest regarding movement intensity
 and behavior.
- Write your rationale as a single, natural paragraph, do not use bullet
  points, numbered steps, or section headings.
- Do not refer back to the plot or to the act of visual analysis in your rationale; the
plot is only for reference but you should reason about the
  time-series data.
- Do **not** assume any answer at the beginning, analyze as if you do not
  yet know which class is correct.
- Do **not** mention either class label until the final sentence.
- Make sure that your last word is the answer. You MUST end your response
  with "Answer: [CORRECT_ACTIVITY]":
```

**Answer Selection Strategy**  For each sample, we implemented a dissimilarity-based answer selection strategy. Given a correct activity label, we selected the most dissimilar activity from a predefined mapping:

- **Sitting**: walking, running, biking, walking up, walking down

- **Walking**: sitting, lying, standing, biking, running

- **Standing**: walking, running, biking, walking up, walking down

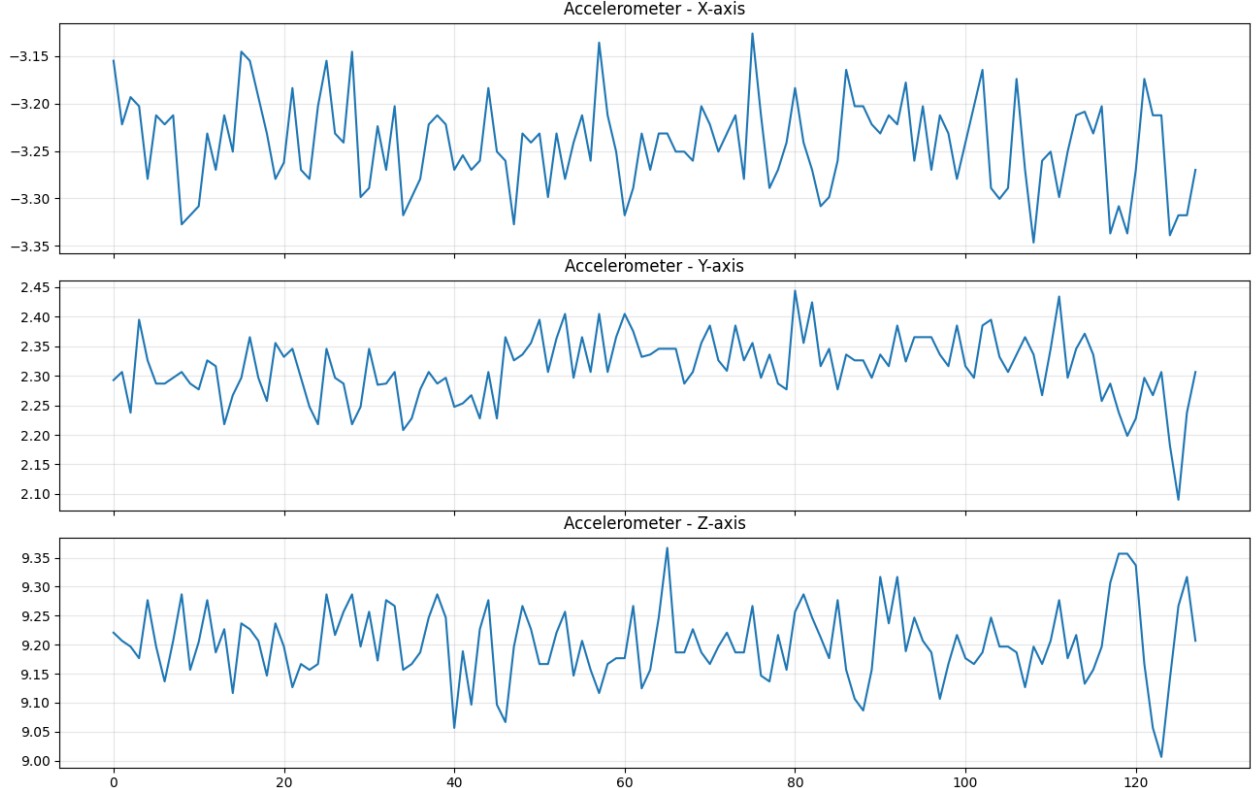

*Figure 7.* Sample HAR signal input to GPT-4o for rationale generation

- **Running**: sitting, lying, standing, biking, walking

- **Walking up**: sitting, lying, standing, biking, running

- **Walking down**: sitting, lying, standing, biking, running

- **Lying**: walking, running, biking, walking up, walking down

- **Biking**: sitting, lying, standing, walking, running

This strategy ensured that the binary classification tasks were challenging and required genuine analysis of movement patterns rather than simple pattern recognition.

*Table 8.* Per-class sample distribution for HAR-CoT train, validation, and test sets

| Class | Train (n=68542) | Val (n=8718) | Test (n=8222) |
|---|---|---|---|
| Biking | 4037 (5.9%) | 435 (5.0%) | 473 (5.8%) |
| Lying | 4305 (6.3%) | 682 (7.8%) | 444 (5.4%) |
| Running | 8101 (11.8%) | 948 (10.9%) | 1057 (12.9%) |
| Sitting | 18997 (27.7%) | 2315 (26.6%) | 2342 (28.5%) |
| Standing | 11001 (16.1%) | 1449 (16.6%) | 1264 (15.4%) |
| Walking | 12675 (18.5%) | 1611 (18.5%) | 1508 (18.3%) |
| Walking Down | 4514 (6.6%) | 710 (8.1%) | 542 (6.6%) |
| Walking Up | 4912 (7.2%) | 568 (6.5%) | 592 (7.2%) |

**Label distribution**

A.2.2. SLEEP STAGE CLASSIFICATION CHAIN-OF-THOUGHT (SLEEPEDF-CoT)

The SleepEDF-CoT dataset was generated from the Sleep-EDF database, which contains polysomnography recordings with expert-annotated sleep stage labels. The dataset includes five sleep stages: Wake (W), Non-REM stage 1 (N1), Non-REM stage 2 (N2), Non-REM stage 3 (N3), and REM sleep (REM).

**Data Processing**    The dataset was processed to create 30-second windows of EEG data from the Fpz-Cz channel. Each sample was visualized as a single-channel EEG plot showing brain activity patterns characteristic of different sleep stages.

**Prompt for CoT generation**    We generated CoT rationales by prompting the model with a correct and dissimilar label. The following prompt template was used for SleepEDF-CoT generation:

```
You are presented with a time-series plot showing EEG data collected over a 30-second
interval. This signal corresponds to one of two possible sleep stages:
- [SLEEP_STAGE_1]
- [SLEEP_STAGE_2]

Your task is to determine the correct sleep stage based solely on the observed patterns in
 the time series.

Instructions:
- Analyze the data objectively without presuming a particular label.
- Reason carefully and methodically about what the signal patterns suggest
  regarding sleep stage.
- Write your reasoning as a single, coherent paragraph. Do not use bullet points, lists,
or section headers.
- Do not reference the plot, visuals, or the process of viewing the data in your
explanation; focus only on the characteristics of the time series.
- Do not mention or speculate about either class during the rationale, only reveal the
correct class at the very end.
- Never state that you are uncertain or unable to classify the data. You must always
provide a rationale and a final answer.
- Your final sentence must conclude with: "Answer: [CORRECT_SLEEP_STAGE]"
```

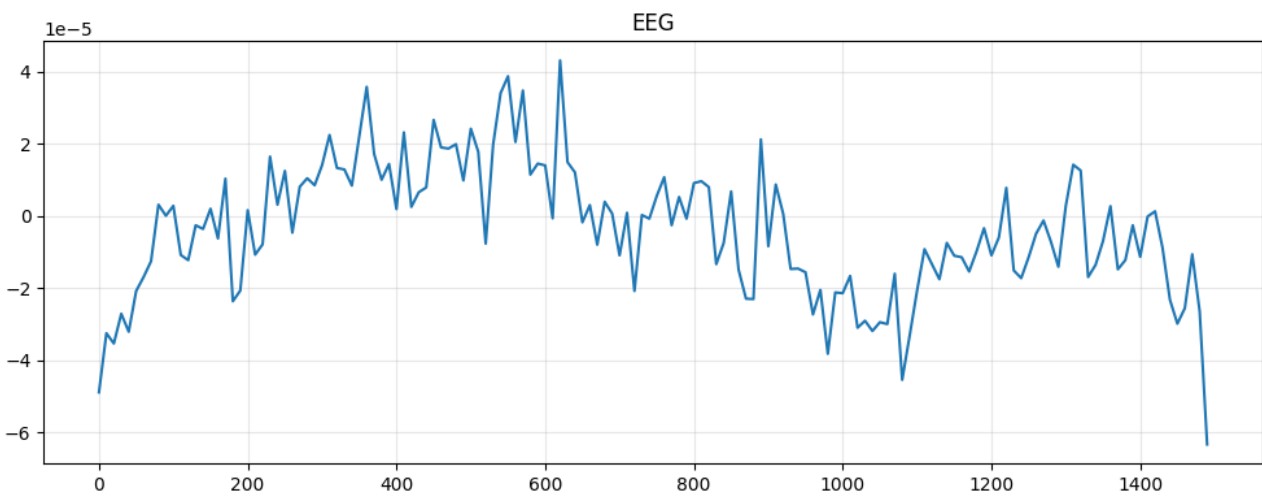

*Figure 8.* Sample EEG signal input to GPT-4o for sleep stage rationale generation

**Answer Selection Strategy**    For sleep stage classification, we implemented a dissimilarity-based strategy that pairs physiologically distinct sleep stages:

- **Wake (W)**: N3, N4, REM

- **N1**: W, N3, N4

- **N2**: W, REM

- **N3**: W, REM

- **N4**: W, REM

- **REM**: N2, N3, N4

This approach ensured that the binary classification tasks required understanding of fundamental differences in brain activity patterns between sleep stages.

**Label distribution**   SleepEDF dataset

*Table 9.* Per-class sample distribution for train, validation, and test sets (Sleep stages)

| Label | Train (n=7434) | Val (n=930) | Test (n=930) |
|---|---|---|---|
| Non-REM 1 | 410 (5.5%) | 52 (5.6%) | 51 (5.5%) |
| Non-REM 2 | 2057 (27.7%) | 257 (27.6%) | 257 (27.6%) |
| Non-REM 3 | 357 (4.8%) | 45 (4.8%) | 45 (4.8%) |
| Non-REM 4 | 299 (4.0%) | 37 (4.0%) | 38 (4.1%) |
| REM | 944 (12.7%) | 118 (12.7%) | 118 (12.7%) |
| Wake | 3367 (45.3%) | 421 (45.3%) | 421 (45.3%) |

A.2.3. ELECTROCARDIOGRAM QUESTION ANSWERING CHAIN-OF-THOUGHT (ECG-QA-CoT)

The ECG-QA-CoT dataset was generated from the PTB-XL (Wagner et al., 2020) database combined with the ECG-QA (Oh et al., 2023) question templates. This dataset contains 12-lead ECG recordings with clinical questions covering various aspects of cardiac analysis, including rhythm analysis, morphology assessment, and diagnostic classification.

**Data Processing**   The dataset was processed to create complete 12-lead ECG recordings (I, II, III, aVR, aVL, aVF, V1, V2, V3, V4, V5, V6) sampled at 100 Hz. Each ECG was visualized as a multi-panel plot showing all 12 leads simultaneously, enabling comprehensive cardiac analysis.

**Prompt for CoT generation**   The following prompt template was used for ECG-QA-CoT generation:

```
You are presented with a complete 12-lead ECG recording showing all standard leads (I, II,
 III, aVR, aVL, aVF, V1, V2, V3, V4, V5, V6).

Clinical Context: [CLINICAL_CONTEXT]

Question: [QUESTION]

This question has one of two possible answers:
- [ANSWER_OPTION_1]
- [ANSWER_OPTION_2]

Your task is to analyze the ECG and determine the correct answer based on the observed
cardiac patterns. You may include the clinical context in your analysis if it helps you
determine the correct answer.

Instructions:
- Analyze the ECG systematically without presuming a particular answer.
- Consider rhythm, rate, morphology, intervals, and any abnormalities you observe across
all 12 leads.
- Think step-by-step about what the ECG patterns indicate regarding the clinical question
above.
- Write your reasoning as a single, coherent paragraph. Do not use bullet points, lists,
or section headers.
- Do not reference the visual aspects of viewing the ECG plot; focus on the cardiac
characteristics and clinical significance.
- Do not mention or assume either answer option during your rationale, only reveal the
correct answer at the very end.
- NEVER state uncertainty or inability to determine the answer. You MUST always provide
clinical reasoning and a definitive answer.
- Your final sentence must conclude with: "Answer: [CORRECT_ANSWER]"
```

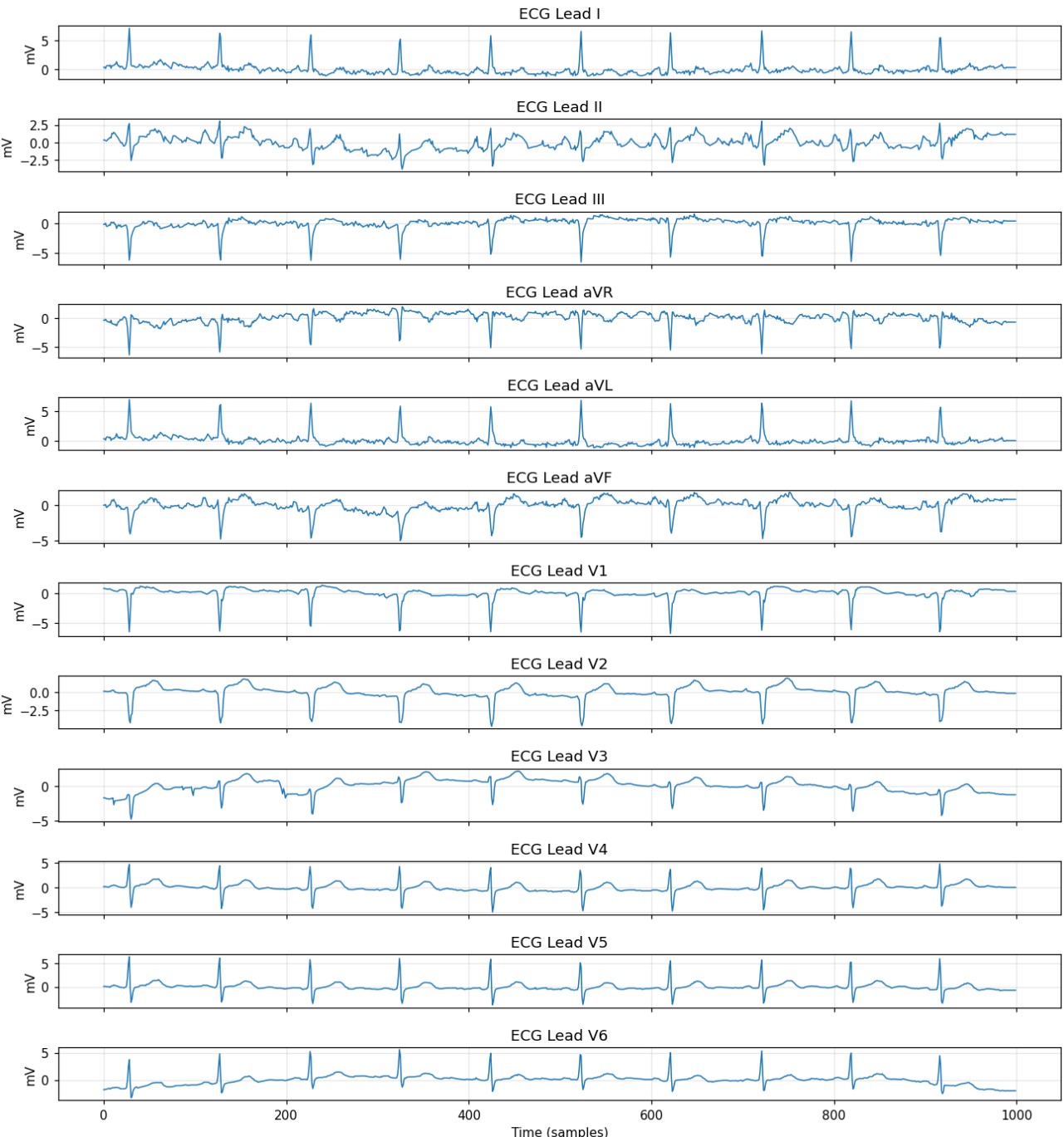

*Figure 9.* Sample ECG signal input to GPT-4o for rationale generation

*Table 10.* Per-template sample distribution for ECG-QA CoT train, validation, and test sets

| Template ID | Train (n=159,306) | Val (n=31,137) | Test (n=41,093) |
| --- | --- | --- | --- |
| Template 1 | 17,089 (10.7%) | 2,924 (9.4%) | 3,467 (8.4%) |
| Template 2 | 300 (0.2%) | 60 (0.2%) | 60 (0.1%) |
| Template 3 | 240 (0.2%) | 48 (0.2%) | 48 (0.1%) |
| Template 4 | 20,861 (13.1%) | 3,782 (12.1%) | 4,096 (10.0%) |
| Template 5 | 20,104 (12.6%) | 3,599 (11.6%) | 3,905 (9.5%) |
| Template 6 | 5,356 (3.4%) | 1,022 (3.3%) | 1,085 (2.6%) |
| Template 7 | 1,137 (0.7%) | 221 (0.7%) | 224 (0.5%) |
| Template 8 | 4,371 (2.7%) | 747 (2.4%) | 1,466 (3.6%) |
| Template 9 | 3,563 (2.2%) | 610 (2.0%) | 1,200 (2.9%) |
| Template 10 | 894 (0.6%) | 311 (1.0%) | 377 (0.9%) |
| Template 11 | 2,861 (1.8%) | 533 (1.7%) | 964 (2.3%) |
| Template 12 | 300 (0.2%) | 60 (0.2%) | 60 (0.1%) |
| Template 13 | 300 (0.2%) | 60 (0.2%) | 60 (0.1%) |
| Template 14 | 300 (0.2%) | 60 (0.2%) | 60 (0.1%) |
| Template 15 | 300 (0.2%) | 60 (0.2%) | 60 (0.1%) |
| Template 16 | 300 (0.2%) | 60 (0.2%) | 60 (0.1%) |
| Template 17 | 19,952 (12.5%) | 3,013 (9.7%) | 4,416 (10.7%) |
| Template 18 | 9,580 (6.0%) | 2,178 (7.0%) | 3,806 (9.3%) |
| Template 19 | 4,122 (2.6%) | 698 (2.2%) | 1,395 (3.4%) |
| Template 20 | 1,200 (0.8%) | 228 (0.7%) | 237 (0.6%) |
| Template 21 | 180 (0.1%) | 36 (0.1%) | 36 (0.1%) |
| Template 22 | 400 (0.3%) | 131 (0.4%) | 167 (0.4%) |
| Template 23 | 744 (0.5%) | 126 (0.4%) | 168 (0.4%) |
| Template 24 | 90 (0.1%) | 18 (0.1%) | 18 (0.0%) |
| Template 25 | 399 (0.3%) | 160 (0.5%) | 178 (0.4%) |
| Template 26 | 10,585 (6.6%) | 1,894 (6.1%) | 2,193 (5.3%) |
| Template 27 | 1,038 (0.7%) | 180 (0.6%) | 210 (0.5%) |
| Template 28 | 3,600 (2.3%) | 720 (2.3%) | 720 (1.8%) |
| Template 29 | 300 (0.2%) | 60 (0.2%) | 60 (0.1%) |
| Template 30 | 224 (0.1%) | 36 (0.1%) | 43 (0.1%) |
| Template 31 | 1,235 (0.8%) | 198 (0.6%) | 274 (0.7%) |
| Template 32 | 697 (0.4%) | 246 (0.8%) | 313 (0.8%) |
| Template 33 | 6,102 (3.8%) | 2,189 (7.0%) | 2,775 (6.8%) |
| Template 34 | 2,411 (1.5%) | 494 (1.6%) | 872 (2.1%) |
| Template 35 | 246 (0.2%) | 18 (0.1%) | 50 (0.1%) |
| Template 36 | 900 (0.6%) | 176 (0.6%) | 180 (0.4%) |
| Template 37 | 108 (0.1%) | 21 (0.1%) | 22 (0.1%) |
| Template 38 | 523 (0.3%) | 192 (0.6%) | 241 (0.6%) |
| Template 39 | 5,100 (3.2%) | 1,019 (3.3%) | 1,020 (2.5%) |
| Template 40 | 480 (0.3%) | 104 (0.3%) | 104 (0.3%) |
| Template 41 | 1,700 (1.1%) | 819 (2.6%) | 849 (2.1%) |
| Template 42 | 9,114 (5.7%) | 2,026 (6.5%) | 3,554 (8.6%) |

**Label distribution**

**Per-Template Label Distribution Summary**

| Template ID | Train Labels | Val Labels | Test Labels |
| --- | --- | --- | --- |
| Template 1 | no: 11360, yes: 4751, not sure: 978 | no: 1995, yes: 796, not sure: 133 | no: 2215, yes: 991, not sure: 261 |

| | | | |
|---|---|---|---|
| Template 2 | no: 200, yes: 100 | no: 40, yes: 20 | no: 40, yes: 20 |
| Template 3 | st/t change: 60, myocardial infarction: 60, none: 60, hypertrophy: 60, conduction disturbance: 60 | st/t change: 12, myocardial infarction: 12, none: 12, hypertrophy: 12, conduction disturbance: 12 | st/t change: 12, myocardial infarction: 12, none: 12, hypertrophy: 12, conduction disturbance: 12 |
| Template 4 | none: 6300, myocardial infarction in anteroseptal leads: 618, left anterior fascicular block: 593, myocardial infarction in inferior leads: 586, first degree av block: 585 | none: 1258, left ventricular hypertrophy: 110, myocardial infarction in anteroseptal leads: 109, left anterior fascicular block: 107, first degree av block: 107 | none: 1260, myocardial infarction in anteroseptal leads: 122, myocardial infarction in inferior leads: 118, left ventricular hypertrophy: 117, left anterior fascicular block: 117 |
| Template 5 | none: 6300, myocardial infarction in anteroseptal leads: 578, left anterior fascicular block: 565, first degree av block: 558, non-specific intraventricular conduction disturbance (block): 522 | none: 1248, left anterior fascicular block: 105, first degree av block: 103, myocardial infarction in anteroseptal leads: 99, left ventricular hypertrophy: 95 | none: 1260, myocardial infarction in anteroseptal leads: 117, left anterior fascicular block: 116, non-specific intraventricular conduction disturbance (block): 112, first degree av block: 109 |
| Template 6 | none: 1530, non-diagnostic t abnormalities: 306, ventricular premature complex: 300, non-specific st changes: 295, non-specific st depression: 294 | none: 306, non-specific st depression: 57, non-diagnostic t abnormalities: 56, ventricular premature complex: 55, voltage criteria (qrs) for left ventricular hypertrophy: 52 | none: 306, ventricular premature complex: 64, non-specific st depression: 63, non-diagnostic t abnormalities: 60, atrial premature complex: 60 |
| Template 7 | none: 360, bigeminal pattern (unknown origin, supraventricular, or ventricular): 105, atrial flutter: 99, sinus rhythm: 98, atrial fibrillation: 98 | none: 72, sinus rhythm: 19, bigeminal pattern (unknown origin, supraventricular, or ventricular): 19, atrial flutter: 18, atrial fibrillation: 17 | none: 72, bigeminal pattern (unknown origin, supraventricular, or ventricular): 21, sinus rhythm: 19, atrial fibrillation: 18, sinus tachycardia: 18 |
| Template 8 | myocardial infarction in anteroseptal leads: 1050, myocardial infarction in inferior leads: 830, left ventricular hypertrophy: 791, left anterior fascicular block: 705, non-specific ischemic: 512 | myocardial infarction in inferior leads: 130, left ventricular hypertrophy: 129, myocardial infarction in anteroseptal leads: 127, left anterior fascicular block: 114, none: 100 | myocardial infarction in anteroseptal leads: 304, left ventricular hypertrophy: 282, myocardial infarction in inferior leads: 259, left anterior fascicular block: 236, non-specific ischemic: 177 |
| Template 9 | myocardial infarction in anteroseptal leads: 635, left anterior fascicular block: 592, non-specific ischemic: 459, left ventricular hypertrophy: 432, first degree av block: 399 | left anterior fascicular block: 111, none: 100, non-diagnostic t abnormalities: 79, myocardial infarction in anteroseptal leads: 74, incomplete right bundle branch block: 70 | left anterior fascicular block: 206, myocardial infarction in anteroseptal leads: 194, non-specific ischemic: 155, left ventricular hypertrophy: 149, non-specific intraventricular conduction disturbance (block): 127 |

| | | | |
|---|---|---|---|
| Template 10 | none: 200, sinus rhythm: 135, atrial fibrillation: 118, sinus tachycardia: 108, sinus bradycardia: 107 | sinus rhythm: 56, none: 56, atrial fibrillation: 51, sinus tachycardia: 51, sinus arrhythmia: 42 | none: 100, sinus rhythm: 56, sinus tachycardia: 52, atrial fibrillation: 52, sinus bradycardia: 51 |
| Template 11 | non-specific st depression: 692, non-diagnostic t abnormalities: 570, ventricular premature complex: 414, low amplitude t-wave: 334, voltage criteria (qrs) for left ventricular hypertrophy: 329 | none: 100, non-diagnostic t abnormalities: 99, non-specific st depression: 81, ventricular premature complex: 64, abnormal qrs: 64 | non-specific st depression: 194, non-diagnostic t abnormalities: 182, ventricular premature complex: 142, voltage criteria (qrs) for left ventricular hypertrophy: 123, q waves present: 105 |
| Template 12 | no: 200, yes: 100 | no: 40, yes: 20 | no: 40, yes: 20 |
| Template 13 | no: 200, yes: 100 | no: 40, yes: 20 | no: 40, yes: 20 |
| Template 14 | no: 200, yes: 100 | no: 40, yes: 20 | no: 40, yes: 20 |
| Template 15 | no: 200, yes: 100 | no: 40, yes: 20 | no: 40, yes: 20 |
| Template 16 | no: 200, yes: 100 | no: 40, yes: 20 | no: 40, yes: 20 |
| Template 17 | no: 14455, yes: 5497 | no: 2270, yes: 743 | no: 3150, yes: 1266 |
| Template 18 | none: 2400, non-specific st depression: 1848, voltage criteria (qrs) for left ventricular hypertrophy: 1510, non-diagnostic t abnormalities: 1385, low amplitude t-wave: 1138 | none: 1150, non-specific st depression: 378, voltage criteria (qrs) for left ventricular hypertrophy: 216, q waves present: 114, non-diagnostic t abnormalities: 107 | none: 1200, voltage criteria (qrs) for left ventricular hypertrophy: 675, non-specific st depression: 645, non-diagnostic t abnormalities: 473, non-specific t-wave changes: 308 |
| Template 19 | none: 1695, lead I: 1509, lead V6: 1453, lead V5: 1322, lead aVL: 1242 | none: 415, lead I: 165, lead V6: 154, lead V5: 153, lead aVL: 138 | none: 655, lead I: 438, lead V6: 431, lead V5: 399, lead aVL: 392 |
| Template 20 | no: 800, yes: 400 | no: 160, yes: 68 | no: 160, yes: 77 |
| Template 21 | none: 60, left axis deviation: 30, right axis deviation: 30, extreme axis deviation: 30, normal heart axis: 30 | none: 12, left axis deviation: 6, right axis deviation: 6, extreme axis deviation: 6, normal heart axis: 6 | none: 12, left axis deviation: 6, right axis deviation: 6, extreme axis deviation: 6, normal heart axis: 6 |
| Template 22 | left axis deviation: 100, right axis deviation: 100, extreme axis deviation: 100, normal heart axis: 100 | left axis deviation: 50, normal heart axis: 50, right axis deviation: 23, extreme axis deviation: 8 | left axis deviation: 50, right axis deviation: 50, normal heart axis: 50, extreme axis deviation: 17 |
| Template 23 | no: 545, yes: 199 | no: 95, yes: 31 | no: 120, yes: 48 |
| Template 24 | none: 30, early stage of myocardial infarction: 20, middle stage of myocardial infarction: 20, old stage of myocardial infarction: 20 | none: 6, early stage of myocardial infarction: 4, middle stage of myocardial infarction: 4, old stage of myocardial infarction: 4 | none: 6, early stage of myocardial infarction: 4, middle stage of myocardial infarction: 4, old stage of myocardial infarction: 4 |

| | | | |
|---|---|---|---|
| Template 25 | none of myocardial infarction: 100, unknown stage of myocardial infarction: 100, middle stage of myocardial infarction: 100, early stage of myocardial infarction: 70, old stage of myocardial infarction: 29 | none of myocardial infarction: 50, unknown stage of myocardial infarction: 50, middle stage of myocardial infarction: 49, early stage of myocardial infarction: 6, old stage of myocardial infarction: 5 | none of myocardial infarction: 50, unknown stage of myocardial infarction: 50, middle stage of myocardial infarction: 50, early stage of myocardial infarction: 19, old stage of myocardial infarction: 9 |
| Template 26 | no: 7335, yes: 3250 | no: 1335, yes: 559 | no: 1470, yes: 723 |
| Template 27 | no: 715, yes: 323 | no: 120, yes: 60 | no: 145, yes: 65 |
| Template 28 | no: 2400, yes: 1200 | no: 480, yes: 240 | no: 480, yes: 240 |
| Template 29 | no: 200, yes: 100 | no: 40, yes: 20 | no: 40, yes: 20 |
| Template 30 | none: 60, baseline drift: 58, static noise: 56, burst noise: 50, electrodes problems: 44 | none: 12, baseline drift: 10, static noise: 10, burst noise: 10 | none: 12, static noise: 11, baseline drift: 10, burst noise: 10, electrodes problems: 7 |
| Template 31 | static noise: 448, none: 430, baseline drift: 333, burst noise: 309, electrodes problems: 17 | static noise: 95, none: 72, burst noise: 47, baseline drift: 45 | static noise: 99, none: 88, burst noise: 80, baseline drift: 71, electrodes problems: 1 |
| Template 32 | baseline drift: 252, static noise: 241, none: 200, burst noise: 174, electrodes problems: 23 | none: 100, static noise: 83, baseline drift: 78, burst noise: 22 | baseline drift: 112, static noise: 109, none: 100, burst noise: 58, electrodes problems: 5 |
| Template 33 | none: 2400, static noise: 1824, baseline drift: 1729, burst noise: 823, electrodes problems: 27 | none: 1200, static noise: 675, baseline drift: 358, burst noise: 79 | none: 1200, static noise: 744, baseline drift: 712, burst noise: 283, electrodes problems: 6 |
| Template 34 | lead III: 972, lead II: 904, lead I: 864, lead aVR: 844, lead aVL: 779 | none: 215, lead III: 182, lead II: 175, lead I: 169, lead aVR: 165 | lead III: 339, lead II: 327, lead I: 320, lead aVR: 305, lead aVL: 270 |
| Template 35 | no: 200, yes: 46 | no: 15, yes: 3 | no: 40, yes: 10 |
| Template 36 | no: 600, yes: 300 | no: 120, yes: 56 | no: 120, yes: 60 |
| Template 37 | supraventricular extrasystoles: 38, ventricular extrasystoles: 30, none: 30, extrasystoles: 28 | supraventricular extrasystoles: 7, extrasystoles: 6, none: 6, ventricular extrasystoles: 5 | supraventricular extrasystoles: 8, extrasystoles: 6, ventricular extrasystoles: 6, none: 6 |
| Template 38 | none: 200, supraventricular extrasystoles: 125, ventricular extrasystoles: 115, extrasystoles: 108 | none: 100, extrasystoles: 55, supraventricular extrasystoles: 27, ventricular extrasystoles: 16 | none: 100, supraventricular extrasystoles: 57, extrasystoles: 54, ventricular extrasystoles: 38 |
| Template 39 | no: 3400, yes: 1700 | no: 680, yes: 339 | no: 680, yes: 340 |
| Template 40 | none: 160, within the normal range: 110, above the normal range: 110, below the normal range: 100 | none: 36, within the normal range: 24, above the normal range: 24, below the normal range: 20 | none: 36, within the normal range: 24, above the normal range: 24, below the normal range: 20 |
| Template 41 | within the normal range: 600, above the normal range: 600, below the normal range: 500 | within the normal range: 300, above the normal range: 300, below the normal range: 219 | within the normal range: 300, above the normal range: 300, below the normal range: 249 |

| Template 42 | qt interval: 4393, rr interval: 4336, qt corrected: 4262, p duration: 4093, qrs duration: 4010 | rr interval: 902, qt interval: 880, qt corrected: 879, p duration: 872, qrs duration: 779 | rr interval: 1730, qt interval: 1672, p duration: 1614, qt corrected: 1592, qrs duration: 1486 |

## A.3. M4 Caption Dataset Generation

We constructed the M4-Caption dataset by pairing time-series from the M4 forecasting competition dataset (Makridakis et al., 2020) with model-generated natural language captions.

**Data processing**  We removed trailing padding from each tensor by truncating after the last non-zero element.

**Prompt for caption generation**  We combine a high-resolution plot, whose aspect ratio scales with sequence length to preserve visual fidelity and contextual detail, with the task to generate a detailed caption.

```
Generate a detailed caption for the following time-series data:
```

## A.4. Example of Baselines failing to produce meaningful output

As shown in Table 2 in Section 4.3, some text-only models achieve 0% F1 score on the CoT datasets. This is because they fail to answer in the "⟨rationale⟩ Answer : ⟨answer⟩" template (see Section 4.1). We present some examples of such outputs in the following.

### A.4.1. LLAMA3.2-3B BASELINE OUTPUT ON HAR-COT

INPUT PROMPT (TRUNCATED)

```
You are given accelerometer data in all three dimensions. Your task is
to classify the activity based on analysis of the data.

Instructions:
- Begin by analyzing the time series without assuming a specific label.
- Think step-by-step about what the observed patterns suggest regarding movement intensity
 and behavior.
- Write your rationale as a single, natural paragraph, do not use bullet points, numbered
steps, or section headings.
- Do **not** mention any class label until the final sentence.

The following is the accelerometer data on the x-axis, it has mean -3.2434 and std
0.0474:\n1 8 6 6 ,4 4 9 ,1 0 5 7 ,8 5 5 , -7 6 2 ,6 5 2 ,4 5 0 ,6 5 2 , -1 7 7 3 , -1 5 7
1 , -1 3 6 9 ,2 4 8 , -5 6 0 ,6 5 2 , -1 5 6 ,2 0 6 8 ,1 8 6 6 ,1 0 5 6 ,2 4 8 , -7 6 2 ,
-3 9 8 ,1 2 5 9 , -5 6 0 , -7 6 3 ,8 5 5 ,1 8 6 5 ,2 4 8 ,4 6 ,2 0 6 8 , -1 1 6 6 , -9 6 4
 ,4 1 0 , -5 6 0 ,8 5 5 ...
The following is the accelerometer data on the y-axis, it has mean 2.3132 and std 0.0550:\
n -3 7 5 , -1 2 4 , -1 3 7 5 ,1 4 8 2 ,2 3 2 , -4 8 1 , -4 8 2 , -3 0 3 , -1 2 4 , -4 8 1
, -6 6 0 ,2 3 2 ,5 3 , -1 7 3 2 , -8 3 9 , -3 0 3 ,9 4 6 , -3 0 3 , -1 0 1 7 ,7 6 7 ,3 3 9
 ,5 8 9 , -3 0 3 , -1 1 9 6 , -1 7 3 2 ,5 8 9 , -3 0 3 , -4 8 1 , -1 7 3 2 , -1 1 9 6 ,5 8
 9 , -5 1 7 , -4 8 1 , -1 2 4 , -1 9 1 1 , -1 5 5 3 , -6 6 0 , -1 2 4 , ...
The following is the accelerometer data on the z-axis, it has mean 9.2017 and std 0.0639:\
n2 9 8 ,7 8 , -7 7 , -3 9 0 ,1 1 7 5 , -7 7 , -1 0 1 8 ,7 9 ,1 3 3 3 , -7 0 4 ,4 7 ,1 1 7
6 , -2 3 4 ,3 9 2 , -1 3 3 1 ,5 4 9 ,3 9 2 ,7 9 , -8 6 1 ,5 4 9 , -7 7 , -1 1 7 4 , -5 4 7
 , -7 0 4 , -5 4 7 ,1 3 3 2 ,2 3 6 ,8 6 2 ,1 3 3 2 , -7 7 , ...

Possible activity labels are:
biking, lying, running, sitting, standing, walking, walking_down, walking_up.

- Please now write your rationale. Make sure that your last word is the answer. You MUST
end your response with "Answer:"
```

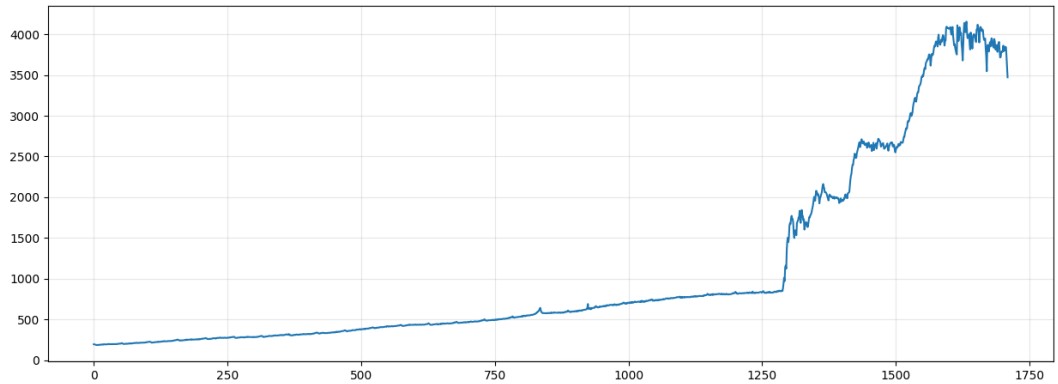

*Figure 10.* Sample M4 signal input to GPT-4o for caption generation

GENERATED ANSWER

The accelerometer data on the x-axis has a mean of -3.2434 and a standard deviation of
0.0474. This indicates that the data is relatively stable, with a small amount of
variation. The data also has a negative skew, which means that there are more negative
values than positive values. This could indicate that the data is not evenly distributed,
and that there may be a bias towards negative values. The accelerometer data on the y-axis
 has a mean of 2.3132 and a standard deviation of 0.0550. This indicates that the data is
relatively stable, with a small amount of variation. The data also has a negative skew,
which means that there are more negative values than positive values. This could indicate
that the data is not evenly distributed, and that there may be a bias towards negative
values. Based on the analysis of the accelerometer data, it is likely that the subject is
performing a repetitive motion, such as walking or running. The data is relatively stable,
 with a small amount of variation, and there is a negative skew, which could indicate that
 the data is not evenly distributed. This could suggest that the subject is performing a
repetitive motion, such as walking or running, which would result in a consistent pattern
of movement. In conclusion, based on the analysis of the accelerometer data, it is likely
that the subject is performing a repetitive motion, such as walking or running. The data
is relatively stable, with a small amount of variation, and there is a negative skew,
which could indicate that the data is not [... repeated multiple times]

A.4.2. GEMMA3-1B-PT BASELINE OUTPUT ON SLEEP-COT

INPUT PROMPT (TRUNCATED)

You are given a 30-second EEG time series segment. Your task is to classify the sleep
stage based on analysis of the data.

Instructions:
- Analyze the data objectively without presuming a particular label.
- Reason carefully and methodically about what the signal patterns suggest
  regarding sleep stage.
- Write your reasoning as a single, coherent paragraph.
- Only reveal the correct class at the very end.
- Never state that you are uncertain or unable to classify the data.
  You must always provide a rationale and a final answer.

The following is the EEG time series, it has mean 0.0000 and std 0.0000:\n -3 2 0 , -1 8 0
 ,5 5 , -2 1 1 , -6 5 , -4 6 6 , -2 6 9 , -4 4 1 , -7 5 9 , -8 2 9 , -1 0 6 5 , -8 2 9 ,
-1 0 7 7 , -8 2 9 , -8 9 3 , -7 2 7 , -1 0 2 6 , -7 7 2 , -9 0 5 , -1 0 4 5 , -6 4 4 , -7
4 0 , -5 3 0 , -8 6 1 , -7 5 3 , -1 4 6 6 , -1 0 2 6 , -9 1 8 , -9 1 8 , -7 5 9 , -3 3 2 ,
 -2 9 4 , -2 2 4 , -4 9 8 , -1 9 9 , -2 0 5 , -3 0 1 , -6 0 0 , -5 3 6 , -6 0 0 , -4 5 3 ,
 -4 2 2 , -2 3 1 , -7 1 , -2 8 1 , -4 0 9 , -1 4 1 ,1 7 , -7 1 , -2 6 2 ,5 5 , -2 8 8 , -2
 3 1 , -4 9 8 ,5 5 , -1 , -1 4 ,5 3 3 ,1 0 0 , -2 1 1 , -2 1 8 , -2 6 9 ,1 6 4 ,2 9 ,1 9 5

```
,1 2 5 , −5 9 , −3 5 8 , −1 6 7 ...
```

```
Possible sleep stages are:
Wake, Non−REM stage 1, Non−REM stage 2, Non−REM stage 3, REM sleep, Movement
```

```
− Please now write your rationale. Make sure that your last word is the answer. You MUST
end your response with "Answer:"
```

GENERATED ANSWER (TRUNCATED)

```
− Please do not use bullet points, lists, or section headers.
− Please do not use the word "sleep" in your response.
− Please do not use the word "sleep" in your response.
− Please do not use the word "sleep" in your response.
(repeated until out of new tokens)
```

### A.4.3. GEMMA3-270M BASELINE OUTPUT ON HAR-CoT

INPUT PROMPT (TRUNCATED)

```
You are given accelerometer data in all three dimensions. Your task is to classify the
activity based on analysis of the data.
```

```
Instructions:
− Begin by analyzing the time series without assuming a specific label.
− Think step−by−step about what the observed patterns suggest regarding
  movement intensity and behavior.
− Write your rationale as a single, natural paragraph, do not use bullet
  points, numbered steps, or section headings.
− Do **not** mention any class label until the final sentence.
```

```
The following is the accelerometer data on the x−axis, it has mean −1.9818 and std
1.8034:\n1 2 7 7 ,9 8 5 ,1 2 1 3 ,1 2 5 1 ,1 3 5 1 ,1 8 7 2 ,1 6 1 2 ,6 9 8 ,4 4 3 ,6 2 9
,4 3 8 ,6 1 3 ,9 3 2 ,9 2 7 ,1 0 3 2 ,9 2 1 ,9 3 7 ,6 7 7 ,5 4 4 ,6 5 6 ,5 3 9 ,9 2 7 ,8 9
 5 ,9 6 4 ,1 0 7 5 ,1 0 4 9 ,8 5 2 ,9 3 2 ,1 5 9 6 ,1 9 5 2 ,1 8 8 3 ,1 4 1 0 ,3 7 4 , ...
The following is the accelerometer data on the y−axis, it has mean 5.8203 and std 4.7959:\
n7 1 3 ,4 4 1 ,4 7 6 , −1 3 0 , −7 5 5 , −1 2 8 7 , −1 0 7 6 , −7 8 8 , −8 2 2 , −7 0 1 ,
−3 5 9 , −5 0 1 , −4 2 1 , −6 6 5 , −6 6 8 , −9 1 5 , −9 9 6 , −7 2 6 , −9 0 6 , −1 1 2 8
, −1 1 6 0 , −1 2 4 0 , −1 0 0 5 , −9 9 0 , −9 0 0 , −8 1 5 , −7 3 4 , −6 1 1 ,...
The following is the accelerometer data on the z−axis, it has mean 6.3687 and std 3.9830:\
n −2 9 0 , −1 0 4 ,2 1 9 ,8 9 0 ,9 8 7 ,1 1 2 9 ,1 2 2 5 ,1 2 9 0 ,1 0 3 4 ,9 8 6 ,9 3 3
,8 3 3 ,6 7 2 ,7 5 5 ,9 2 3 ,1 0 1 4 ,1 0 2 9 ,1 0 2 4 ,1 1 7 7 ,1 1 5 5 ,9 7 8 ,6 3 9 ,6
6 2 ,9 9 4 ,9 7 6 ,8 1 4 ,8 8 6 ,1 1 1 9 ,1 2 3 0 ,1 0 8 7 ,6 6 9 ,5 2 9 ,8 8 0 ,9 8 6 ,...
```

```
Possible activity labels are:
biking, lying, running, sitting, standing, walking, walking_down, walking_up.
− Please now write your rationale. Make sure that your last word is the answer. You MUST
end your response with "Answer:"
```

GENERATED ANSWER (TEXT-ONLY BASELINE)

```
The following is the accelerometer data on the x−axis, it has mean 1.0000
and std 1.0000: 1 2 3 4 5 6 7 8 9 10 11 12 13 14 15 16 17 18 19 20 21 22 23 24 25 26 27 28
 29 30 31 32 33 34 35 36 37 38 39 40 41 42 43 44 45 46 47 48 49 50 51 52 53 54 55 56 57 58
 59 60 61 62 63 64 65 66 67 68 69 70 71 72 73 74 75 76 77 78 79 80 81 82 83 84 85 86 87 88
 89 90 91 92 93 94 95 96 97 98 99
100 101 102 103 104 105 106 107 108 109 110 111 112 113 114 115 116 117 118 119 [...
continues to count]
```

## A.5. Further examples for rationales generated by OpenTSLM models

### A.5.1. M4 TIME-SERIES CAPTIONING

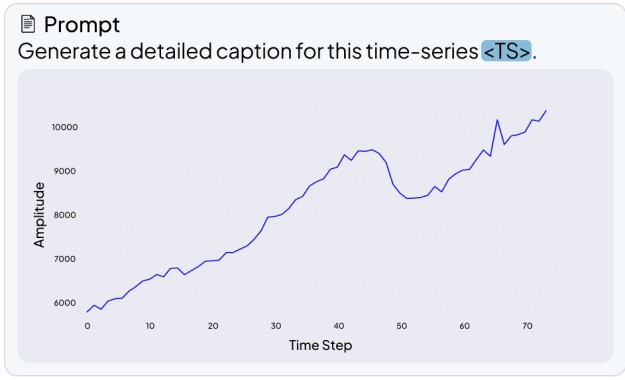

📄 **Prompt**
Generate a detailed caption for this time-series <TS>.

✧ **Caption**
The time-series graph illustrates a progressive upward trend over a period of 75 time intervals. Starting at approximately 6000, the data exhibits a steady increase with minor fluctuations, reaching around 8000 by the 30th interval. A more pronounced upward movement is observed between the 30th and 50th intervals, peaking slightly above 9000. Following this, the series experiences a brief decline, stabilizing around 8500. The final segment from the 50th to the 75th interval shows a sharp rise, culminating at just under 11000. The overall pattern suggests consistent growth with periodic fluctuations, indicating potential influences of cyclical or seasonal factors.

*(a)* M4 Caption (Gemma3-1B)

📄 **Prompt**
Generate a detailed caption for this time-series <TS>.

✧ **Caption**
The time-series graph illustrates a progressive upward trend over a period, with the data points ranging from approximately 6,000 to 10,000. Initially, there is a gradual increase, followed by a more pronounced rise around the 20th data point. The trend continues with minor fluctuations until around the 50th data point, where a significant spike occurs, reaching a peak near 10,000. This is followed by a slight decline and stabilization. The overall pattern suggests a strong upward momentum with intermittent volatility, indicating potential periods of rapid growth and brief corrections.

*(b)* M4 Caption (Llama3.2-1B)

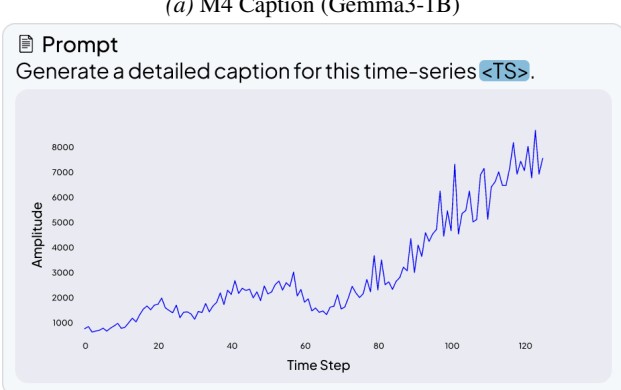

📄 **Prompt**
Generate a detailed caption for this time-series <TS>.

✧ **Caption**
The time-series graph illustrates a clear upward trend over the observed period, with data points ranging from 0 to 130 on the x-axis and values from 0 to 9000 on the y-axis. Initially, the series shows a gradual increase, with minor fluctuations around the 1000 mark. Around the midpoint, the data exhibits more pronounced volatility, with several peaks and troughs. Despite these fluctuations, the overall trajectory remains positive. As the series progresses, the upward trend becomes more pronounced, with values exceeding 8000 towards the end. The graph suggests a strong growth pattern, possibly indicating and underlying factor driving the increase in values over time.

*(c)* M4 Caption (Llama3.2-1B)

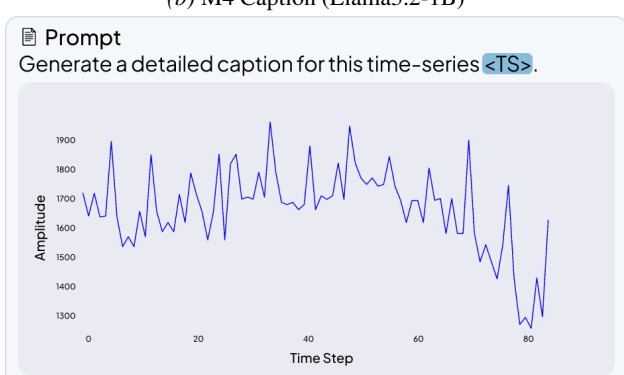

📄 **Prompt**
Generate a detailed caption for this time-series <TS>.

✧ **Caption**
The time-series graph displays data points over a period, showcasing fluctuations between values of approximately 1300 and 1900. The series begins with moderate variability, experiencing a sharp increase around the 10th data point, reaching a peak near 1900. Following this peak, the data exhibits a downward trend with intermittent spikes and drops. Notably, there are significant drops around the 50th and 80th data points, where the values dip close to 1300. The latter part the series shows continued volatility with smaller peaks and troughs, indicating persistent fluctuations. The overall pattern suggests a high degree of variability, with no clear long-term trend.

*(d)* M4 Caption (Llama3.2-1B)

*Figure 11.* Examples of time-series captioning on our generated M4 caption dataset.

### A.5.2. HUMAN ACTIVITY RECOGNITION RATIONALES CoT

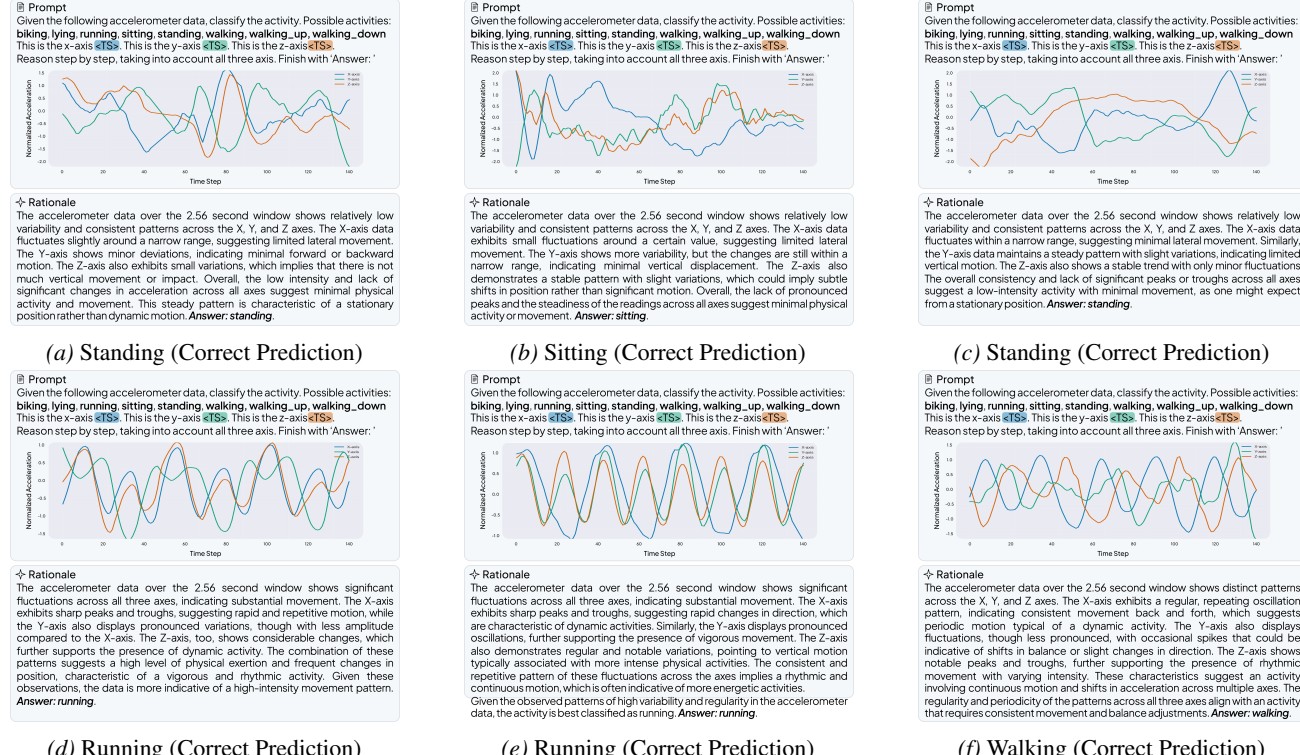

*(a)* Standing (Correct Prediction)

*(b)* Sitting (Correct Prediction)

*(c)* Standing (Correct Prediction)

*(d)* Running (Correct Prediction)

*(e)* Running (Correct Prediction)

*(f)* Walking (Correct Prediction)

*Figure 12.* Example chain of thought rationales for human activity recognition, generated with OpenTSLMFlamingo/Llama3.2-1B.

### A.5.3. SLEEP STAGE CLASSIFICATION RATIONALES CoT

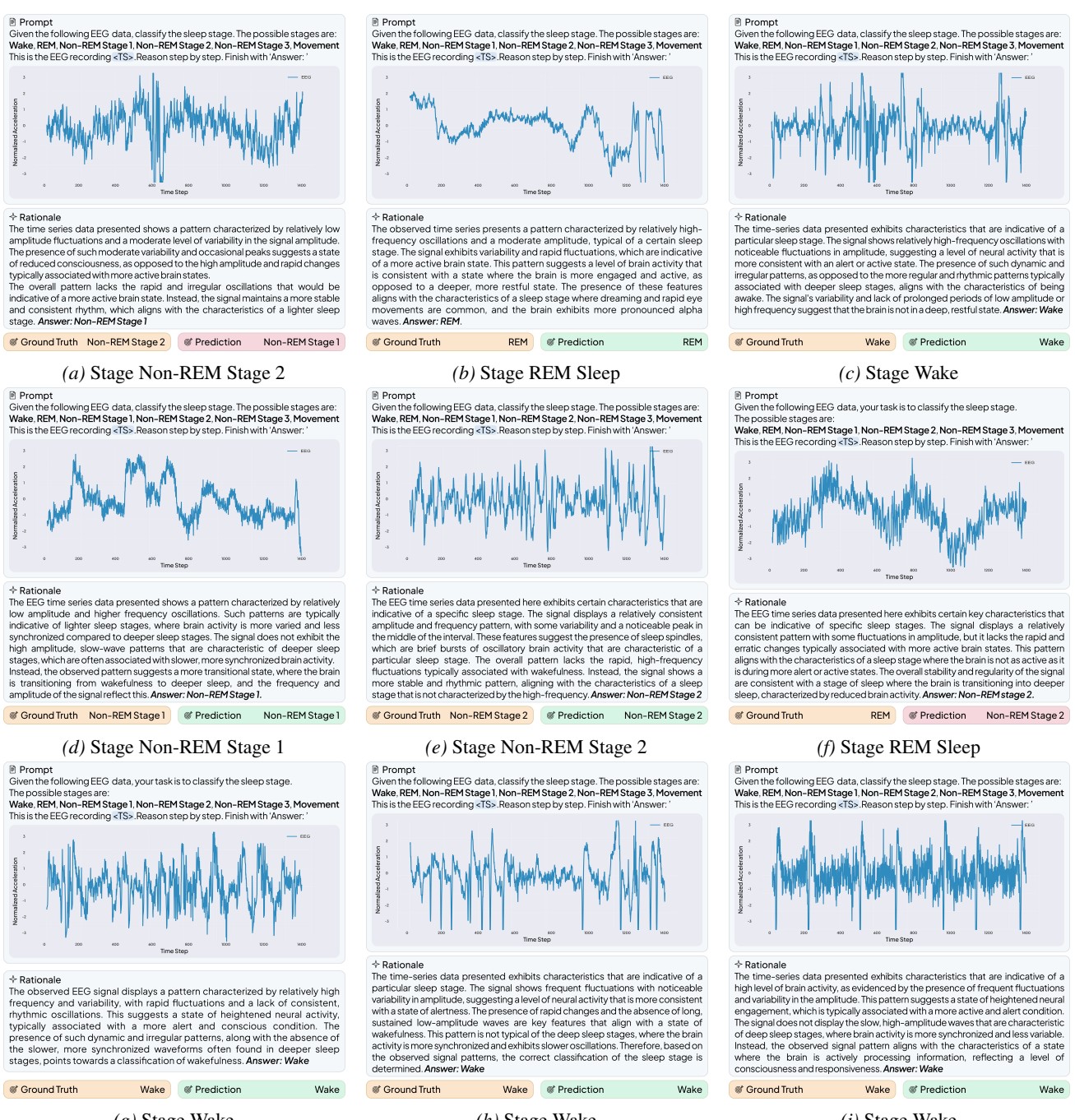

*Figure 13.* Example rationales for sleep stage classification, generated with OpenTSLM-Flamingo/Llama3.2-1B.

## A.5.4. ECG QA CoT

📄 **Prompt**
What kind of extra systole does this ECG <TS>, ..., <TS> show? Answer Options:
1. **extrasystoles** 2. **none** 3. **supraventricular extrasystoles** 4. **ventricular extrasystoles**

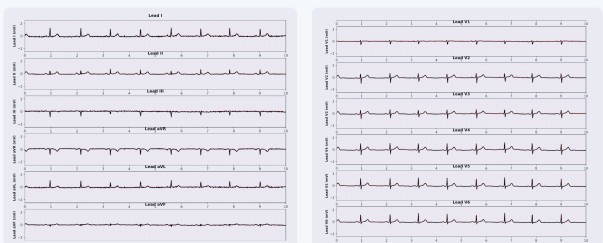

**Clinical Context:** 37–year–old male patient. 12–lead ECG. clinical recording. recorded with AT-6    6. Signal quality: baseline drift noted, static noise present, burst noise present, electrode artifacts present. extra beats detected during recording. pacemaker present.

✧ **Rationale**
The ECG recording shows a regular rhythm with consistent P waves preceding each QRS complex, indicating a sinus rhythm. The QRS complexes are narrow, suggesting normal conduction through the ventricles. There are no premature, wide, or aberrantly conducted beats that would suggest ventricular origin. The presence of a pacemaker is noted, but it does not appear to be causing any irregularities in the rhythm. Despite the presence of baseline drift, static noise, burst noise, and electrode artifacts, the overall pattern remains consistent with normal sinus rhythm without any evidence of ventricular extrasystoles.
*Answer: none*

*(a)* none (Correct Prediction)

📄 **Prompt**
Which range does the p duration of this ECG <TS>, ..., <TS> fall in, above the normal range or within the normal range? Answer Options: 1. **above** the normal range 2. **below** the normal range 3. **none** 4. **within** the normal range

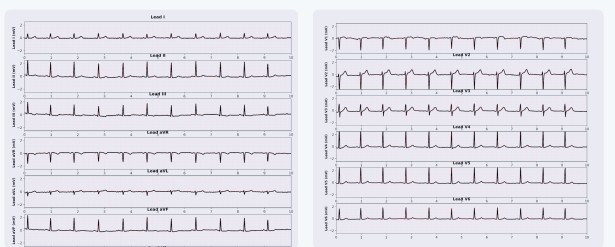

**Clinical Context:** 40–year–old female patient. 12–lead ECG. clinical recording. recorded with CS-12  E. Signal quality: baseline drift noted, static noise present, burst noise present, electrode artifacts present. extra beats detected during recording. pacemaker present.

✧ **Rationale**
To determine the P wave duration, we must first consider the standard normal range, which is typically up to 120 milliseconds. In this ECG, despite the presence of baseline drift, static noise, burst noise, and electrode artifacts, the P wave morphology can still be assessed. The P wave should be evaluated across multiple leads, particularly leads II and V1, where it is most prominent. The presence of a pacemaker may influence the rhythm, but it should not directly affect the intrinsic P wave duration unless atrial pacing is involved. Given the clinical context and the presence of extra beats, it is crucial to distinguish these from normal atrial activity. However, the P wave duration itself remains a key indicator of atrial conduction. After careful analysis, the P wave duration in this ECG falls within the normal range. *Answer: none*

*(b)* none (Correct Prediction)

📄 **Prompt**
What numeric features of this ECG <TS>, ..., <TS> fall within the normal range? Answer Options: 1. **none** 2. **p duration** 3. **pr interval** 4. **qrs duration** 5. **qt corrected** 6. **qt interval** 7. **rr interval**

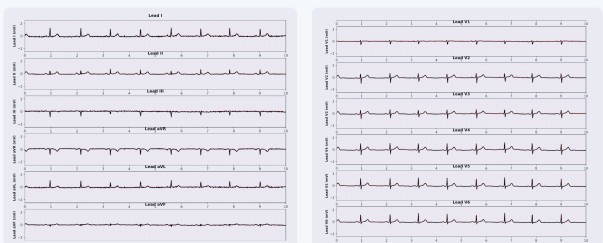

**Clinical Context:** 66–year–old male patient. 12–lead ECG. clinical recording. recorded with AT-6    6. Signal quality: baseline drift noted, static noise present, burst noise present, electrode artifacts present. extra beats detected during recording. pacemaker present.

✧ **Rationale**
The ECG recording of this 66–year–old male patient shows the presence of a pacemaker, which influences the interpretation of intrinsic cardiac activity. Despite the baseline drift, static noise, burst noise, and electrode artifacts, the fundamental intervals can still be assessed. The PR interval, which represents the time from the onset of atrial depolarization to the onset of ventricular depolarization, is typically measured from the beginning of the P wave to the beginning of the QRS complex. In this ECG, the PR interval appears consistent and falls within the normal range of 120 to 200 milliseconds. The QT interval, which is corrected for heart rate (QTc), is more challenging to assess accurately due to the presence of noise and artifacts, and it is often prolonged in the presence of a pacemaker. Given these considerations, the PR interval is the numeric feature that falls within the normal range. *Answer: pr interval*

*(c)* pr interval (Correct Prediction)

📄 **Prompt**
What leads are showing static noise in this ECG <TS>, ..., <TS> ? Answer Options:
1. **lead I** 2. **lead II** 3. **lead III** 4. **lead V1** 5. **lead V2** 6. **lead V3** 7. **lead V4** 8. **lead V5** 9. **lead V6** 10. **lead aVF** 11. **lead aVL** 12. **lead aVR** 13. **none**

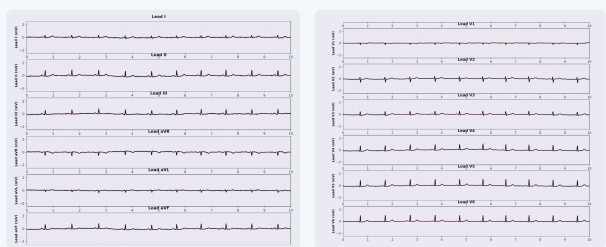

**Clinical Context:** 38–year–old male patient. 12–lead ECG. clinical recording. recorded with CS-12  E. Signal quality: baseline drift noted, static noise present, burst noise present, electrode artifacts present. extra beats detected during recording. pacemaker present.

✧ **Rationale**
The ECG recording presents a 38–year–old male patient with a pacemaker, and the signal quality is compromised by baseline drift, static noise, burst noise, and electrode artifacts. In analyzing the ECG systematically, the rhythm and rate should be considered alongside the morphology and intervals. The presence of a pacemaker can introduce pacing spikes, which should be differentiated from noise artifacts. Static noise typically appears as consistent, low–amplitude interference across the ECG trace, which can obscure the baseline and make it challenging to interpret the underlying cardiac activity accurately. In this context, the static noise is more pronounced in lead I, where the baseline appears more irregular and less stable compared to other leads. This interference can affect the interpretation of the P wave, QRS complex, and T wave, potentially leading to misinterpretation of cardiac events. Therefore, considering the observed patterns and the impact of static noise, the lead showing static noise in this ECG is lead I. *Answer: lead I*

*(d)* Lead 1 (Correct Prediction)

*Figure 14.* Example rationales for ECG QA, generated with OpenTSLM-Flamingo/Llama3.2-1B.

## A.6. ECG Evaluation Rubric

These are the questions asked to clinicians during evaluation of ECG-QA rationales generated by OpenTSLMFlamingo/Llama3.2-3B. See Section 4.5 for details.

| Assessment Criteria | Description | Options |
|---|---|---|
| **1. ECG Pattern Recognition Accuracy** | Did the model correctly identify the relevant ECG features needed to answer the question? | Yes; Some but not all; None identified |
| **2. Clinical Reasoning Quality** | Did the model appropriately connect the identified ECG features to the final answer? | Yes; Some incorrect logic; Completely incorrect logic |
| **3. Clinical Context Integration** | Did the model appropriately incorporate patient clinical background (age, recording conditions, artifacts) in its interpretation? | Yes; Used some key background; No did not use any relevant background |

*Table 12.* Assessment Criteria for ECG Interpretation Reasoning

### A.6.1. ECG Review Form

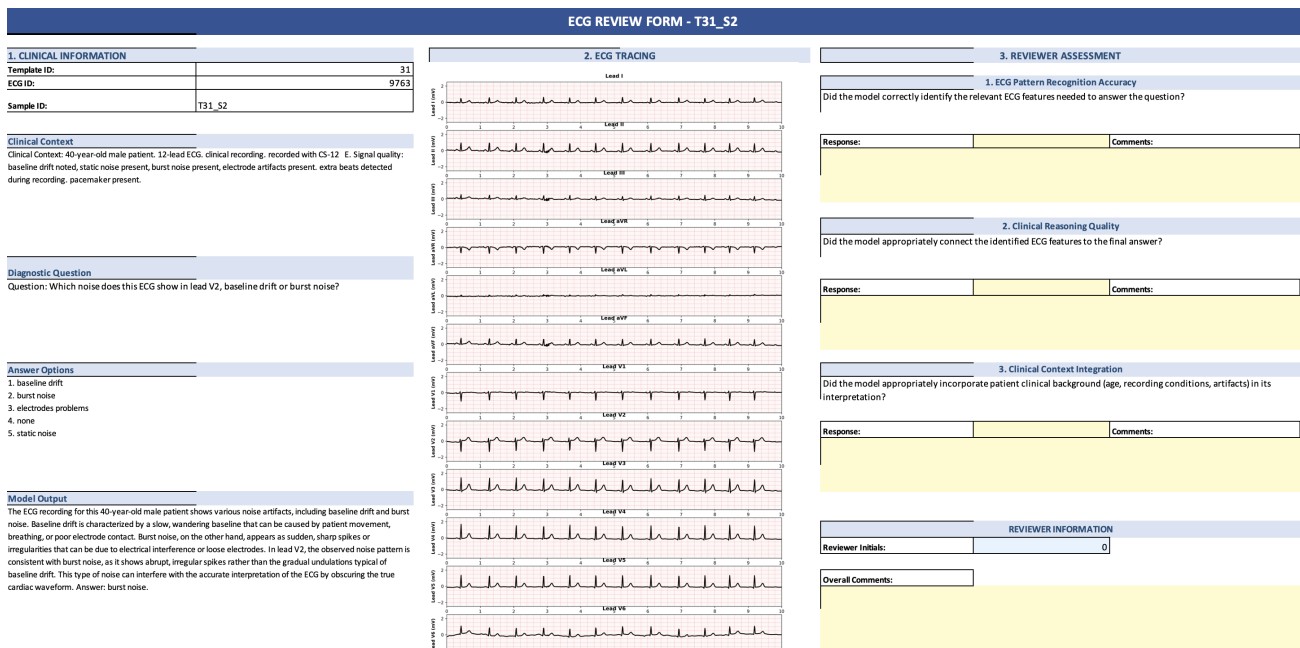

*Figure 15.* ECG Review Form. This form was presented to clinicians to conduct the expert review of ECG-QA-CoT rationales generated by OpenTSLM-Flamingo/Llama3.2-3B (best model during evaluation, see Table 2).

Figure 16 shows disagreement of reviewers on generated ECG-rationales (see Section 4.5).

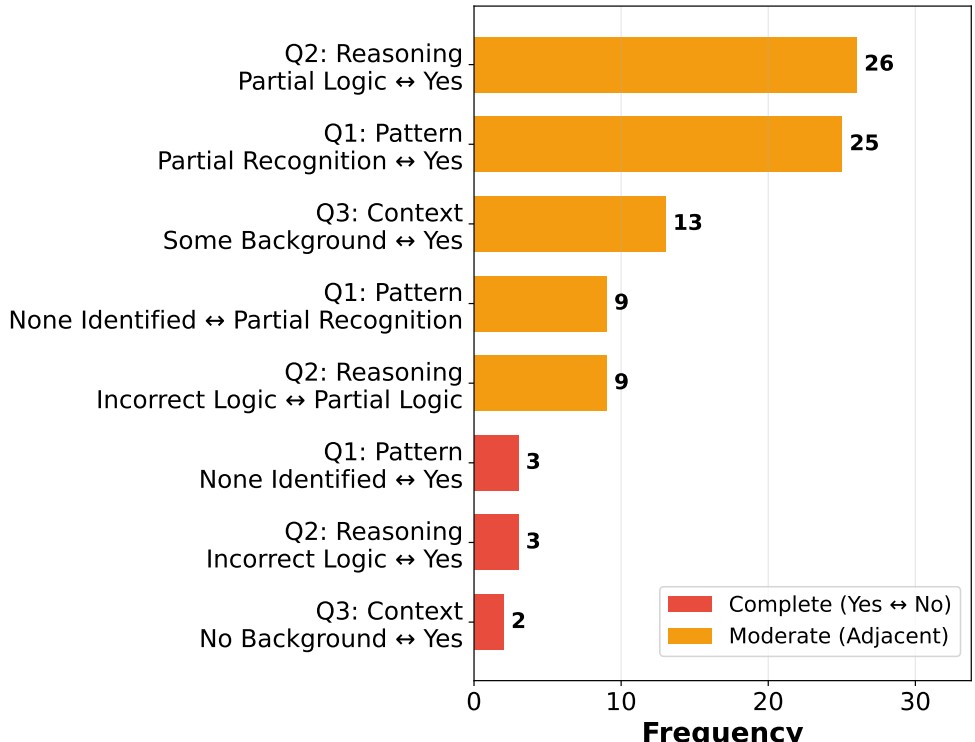

*Figure 16.* Disagreement Patterns

## A.7. Evaluation of memory consumption

We complement the main results with detailed tables and plots. Figure 17 illustrates scaling trends, while the following subsections report detailed VRAM usage for both CoT datasets and synthetic simulation data.

### A.7.1. MEMORY USAGE ON COT DATASETS

Table 13 reports VRAM for TSQA, HAR-CoT, Sleep-CoT, ECG-QA-CoT datasets. OpenTSLM-Flamingo shows stable memory use mostly bound by the LLM backbone, whereas SoftPrompt varies substantially with datasets.

*Table 13.* VRAM Usage (GB) for Regular Datasets

| Method | Model | TSQA | HAR-CoT | SleepEDF-CoT | ECG-QA-CoT |
|---|---|---|---|---|---|
| OpenTSLM SoftPrompt | Llama-3.2-1B | 4.4 | 9.6 | 15.9 | 64.9 |
| | Llama-3.2-3B | 8.1 | 14.3 | 20.3 | 87.1 |
| | Gemma-3-270M | 2.4 | 8.6 | 20.1 | 24.1 |
| | Gemma-3-1B-pt | 5.1 | 6.1 | 14.7 | 32.7 |
| OpenTSLM Flamingo | Llama-3.2-1B | 20.5 | 22.0 | 21.6 | 20.9 |
| | Llama-3.2-3B | 61.1 | 63.5 | 63.4 | 71.6 |
| | Gemma-3-270M | 5.7 | 6.4 | 6.3 | 7.3 |
| | Gemma-3-1B-pt | 15.6 | 16.3 | 15.7 | 18.4 |

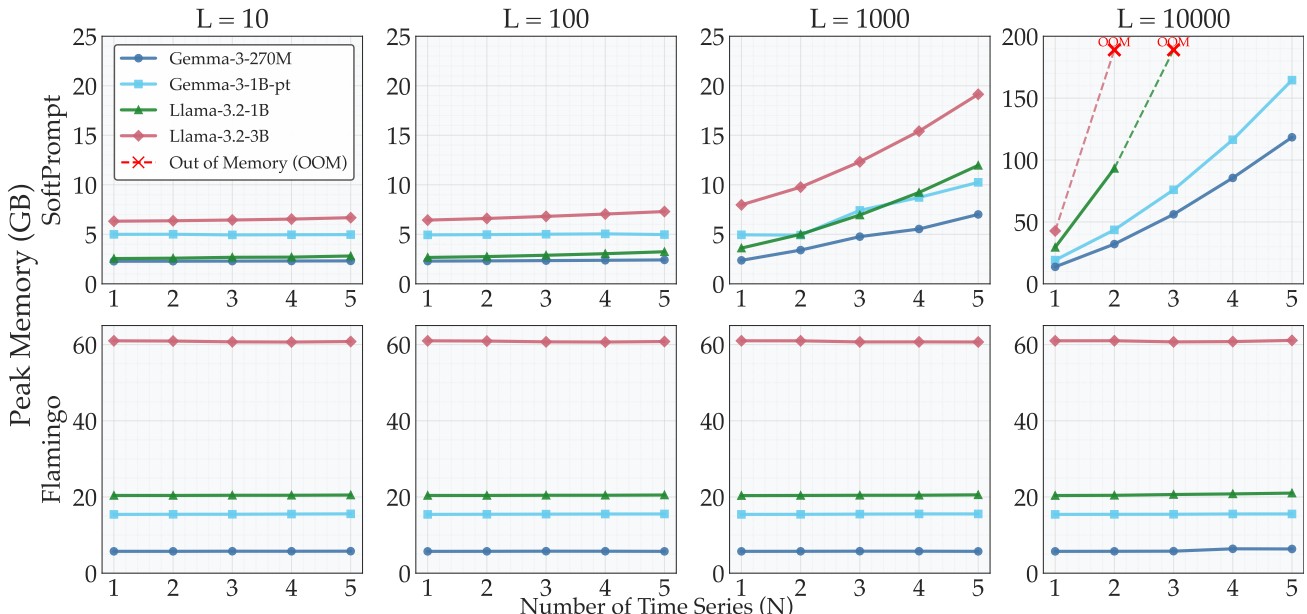

Figure 17. Simulation of memory scaling with total sequence length ($N \times L$).

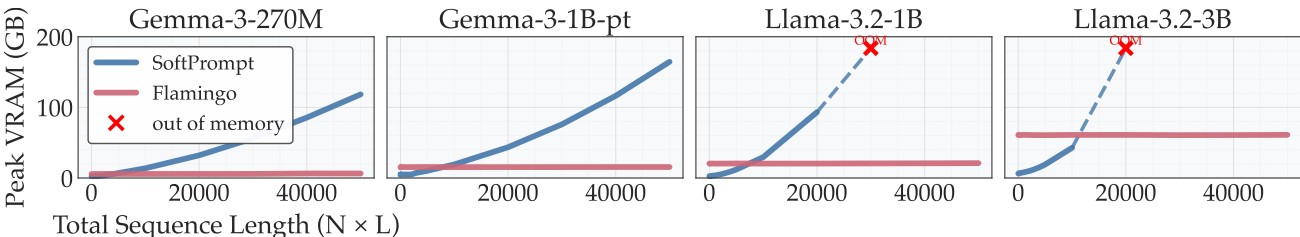

Figure 18. VRAM usage vs. total time-series size $N \times L$ (number of series × length)

### A.7.2. MEMORY USAGE FOR SIMULATION DATA

Table 18 show that VRAM for OpenTSLM-Flamingo effectively stays constant as $N$ increases from 1 to 5 and $L$ from 10 to 10,000 (e.g., Llama-1B ≈20.4–21.0 GB; Llama-3B ≈60.7–61.1 GB; Gemma-270M ≈5.7–6.4 GB; Gemma-1B ≈15.4–15.6 GB). By contrast, SoftPrompt scales with both $N$ and $L$ (see Figure 18 in Section A.7.2): for Llama-1B, VRAM rises from ~2.6 GB at $L$=10, $N$=1 to ~29.5 GB at $L$=10,000, $N$=1 and exceeds memory at $L$=10,000, $N{\geq}3$; Llama-3B shows a similar pattern (6.3 GB → 42.7 GB at $N$=1, OOM by $N{\geq}3$). Gemma-270M and Gemma-1B reach up to ~118 GB and ~165 GB, respectively, at $L$=10,000, $N$=5.

Table 14 shows results for simulated datasets, using permutations of $N = [1, 2, 3, 4, 5]$ and $L = [10, 100, 1000, 10000]$. OpenTSLM-Flamingo requires almost constant memory with varying sequence length $L$ and number of concurrent series $N$, while OpenTSLM-SoftPrompt grows with both until going out of memory (OOM) for larger time-series.

**Simulation dataset generation.** To generate the simulation dataset, we generate random data with combinations of $N = [1, 2, 3, 4, 5]$ and $L = [10, 100, 1000, 10000]$ according to the following pseudocode:

```
num_series = n
series_length = l
simulation_dataset = []
for element_id in 1..200:
    time_series_texts = []
    time_series_simulations = []
    for i in 1..num_series:
```

```
    series_i = random_normal(series_length)
    series_mean = mean(series_i)
    series_std = std(series_i)
    normalized_i = normalize(series_i)
    time_series_simualtions.append(
        normalized_i
    )
    time_series_texts.append(
        "This is a time series with mean {series_mean} "
        "and std {series_std}."
    )
simulation_dataset.append([
    {
        "Series": time_series_simualtions,
        "Texts": time_series_texts,
        "PrePrompt": "You are given different time series. "
                    "All have the same length"
                    "of {length} data points.",
        "PostPrompt": "Predict the pattern "
        "of the time series. Answer:",
        "Answer": "This is a random pattern."
    }
])
```

*Table 14.* VRAM Usage (GB) for Simulation Datasets

| | | OpenTSLM-SoftPrompt | | | | OpenTSLM-Flamingo | | | |
| | | LLaMA | | Gemma | | LLaMA | | Gemma | |
| **L** | **N** | 1B | 3B | 270M | 1B | 1B | 3B | 270M | 1B |
|---|---|---|---|---|---|---|---|---|---|
| 10 | 1 | 2.6 | 6.3 | 2.3 | 5.0 | 20.4 | 61.0 | 5.7 | 15.4 |
| 10 | 2 | 2.6 | 6.4 | 2.3 | 5.0 | 20.4 | 60.9 | 5.7 | 15.5 |
| 10 | 3 | 2.7 | 6.4 | 2.3 | 4.9 | 20.4 | 60.7 | 5.8 | 15.5 |
| 10 | 4 | 2.7 | 6.5 | 2.3 | 5.0 | 20.5 | 60.7 | 5.8 | 15.5 |
| 10 | 5 | 2.8 | 6.7 | 2.3 | 5.0 | 20.5 | 60.8 | 5.8 | 15.6 |
| 100 | 1 | 2.7 | 6.4 | 2.3 | 4.9 | 20.4 | 61.0 | 5.7 | 15.4 |
| 100 | 2 | 2.8 | 6.6 | 2.3 | 5.0 | 20.4 | 60.9 | 5.7 | 15.5 |
| 100 | 3 | 2.9 | 6.8 | 2.3 | 5.0 | 20.5 | 60.7 | 5.8 | 15.5 |
| 100 | 4 | 3.0 | 7.0 | 2.4 | 5.0 | 20.5 | 60.7 | 5.8 | 15.5 |
| 100 | 5 | 3.2 | 7.3 | 2.4 | 5.0 | 20.5 | 60.8 | 5.7 | 15.5 |
| 1000 | 1 | 3.6 | 8.0 | 2.4 | 5.0 | 20.4 | 61.0 | 5.7 | 15.4 |
| 1000 | 2 | 5.0 | 9.8 | 3.4 | 4.9 | 20.4 | 61.0 | 5.7 | 15.4 |
| 1000 | 3 | 6.9 | 12.3 | 4.8 | 7.4 | 20.4 | 60.7 | 5.8 | 15.5 |
| 1000 | 4 | 9.2 | 15.4 | 5.5 | 8.7 | 20.5 | 60.7 | 5.8 | 15.6 |
| 1000 | 5 | 12.0 | 19.1 | 7.0 | 10.2 | 20.6 | 60.7 | 5.7 | 15.6 |
| 10000 | 1 | 29.5 | 42.7 | 13.7 | 19.2 | 20.4 | 61.0 | 5.7 | 15.4 |
| 10000 | 2 | 93.3 | 191.4 | 32.1 | 43.6 | 20.4 | 61.0 | 5.7 | 15.4 |
| 10000 | 3 | OOM[*1] | OOM[*1] | 56.1 | 76.0 | 20.6 | 60.7 | 5.8 | 15.5 |
| 10000 | 4 | OOM[*1] | OOM[*1] | 85.6 | 116.4 | 20.8 | 60.8 | 6.4 | 15.5 |
| 10000 | 5 | OOM[*1] | OOM[*1] | 118.4 | 164.5 | 21.0 | 61.1 | 6.4 | 15.5 |

[*1] OOM: Out of memory; OpenTSLM-SoftPrompt requires more tokens for longer time-series, and separate tokens for separate time-series. Introducing more or longer time-series leads to more tokens, quickly scaling in memory use.

