# OpenReview forum: "OpenTSLM: Time-Series Language Models for Reasoning over Multivariate Medical Text- and Time-Series Data"
_ICML.cc/2026/Conference — ICML 2026 regular_

### Official Review · Reviewer_XwpS · 2026-03-01

**Soundness:** 3
**Presentation:** 3
**Significance:** 2
**Originality:** 3
**Overall Recommendation:** 4
**Confidence:** 3

**Summary:**

The authors attempt to analyze a critical issue that large language models (LLMs), despite their strong multimodal interpretation capabilities, are inherently limited in natively processing time-series data, especially for multivariate medical data that combines temporal signals and text. The manuscript's main area consists of the design, implementation, and comprehensive evaluation of Time-Series Language Models (TSLMs) that integrate time-series as a native modality into pretrained LLMs, with a focus on medical time-series reasoning tasks. Specifically, the authors propose OpenTSLM, a family of TSLMs with two variants: OpenTSLM-SoftPrompt based on soft prompting, and OpenTSLM-Flamingo built on a gated cross-attention mechanism inspired by vision-language models. To support model training and evaluation, the authors construct three new chain-of-thought (CoT) datasets for medical time-series tasks: HAR-CoT for human activity recognition, Sleep-CoT for sleep staging, and ECG-QA-CoT for electrocardiogram question answering. Through extensive experiments, the authors demonstrate that OpenTSLM models consistently outperform text-only LLM baselines, image-based multimodal baselines, and GPT-4o across all tasks. They further validate the model's clinical reasoning ability via expert evaluation with cardiologists, analyze the memory scalability of the two architecture variants, and show that combining OpenTSLM with pretrained time-series foundation models (TSFMs) like Chronos-2 can further boost performance. All code, datasets, and models are released as open-source resources to facilitate follow-up research.

**Compliance With Llm Reviewing Policy:**

Affirmed.

**Final Justification:**

I will maintain my current score.

**Key Questions For Authors:**

1. The rationales of the three CoT datasets are all generated by GPT-4o, which has been proven to perform poorly on time-series reasoning tasks in your experiments. Have you conducted large-scale or full manual quality verification on the generated CoT rationales, and are there quantitative indicators (e.g., human-annotated accuracy, logical integrity) to evaluate their quality?
2. The manuscript does not include direct quantitative comparisons with state-of-the-art time-series LLM methods such as ITFormer, InstructTime, and Time2Lang mentioned in the related work. Can you supplement the experimental results of these SOTA methods on the three CoT datasets proposed in this paper?
3. All experiments in the manuscript are focused on medical time-series tasks. Have you verified the generalization ability of OpenTSLM on general time-series tasks outside the medical field (e.g., financial time-series analysis, industrial anomaly detection)?
4. The manuscript only compares the performance and memory overhead of the SoftPrompt and Flamingo architectures through experiments, without theoretical analysis. Can you supplement the theoretical analysis of the two fusion architectures in terms of expressive power, temporal modeling capability, and computational complexity?

**Limitations:**

yes

**Strengths And Weaknesses:**

## Strengths
1. The work systematically compares two mainstream time-series-LLM fusion paradigms (soft prompting and cross-attention) with rigorous implementation, and evaluates the models from multiple dimensions including quantitative task performance, memory consumption scaling, and clinical expert qualitative assessment. All core claims are supported by detailed and reproducible experiments, with clear ablation studies on the integration of pretrained TSFMs, and the experimental design fully aligns with the research objectives.
2. The work addresses the key pain point of translating longitudinal medical sensor data into interpretable, actionable insights via natural language interaction. The three newly constructed CoT medical time-series datasets fill the gap of high-quality reasoning-focused benchmarks for time-series LLMs in the clinical domain. The full open-source release of the framework, models, and datasets provides a solid foundation for subsequent research on time-series language models, especially in digital health applications.
3. The work introduce Flamingo-style gated cross-attention into time-series-LLM fusion, and explicitly verifies that this design solves the core memory explosion problem of soft prompting methods, which scales poorly with increasing time-series length and quantity. Meanwhile, the framework preserves the native text generation and reasoning capabilities of pretrained LLMs, which is a clear improvement over prior works that discard generation capabilities via fixed classification heads.

## Weaknesses
1. The manuscript only provides implementation details for the two fusion architectures (Sections 3.2 and 3.3), but lacks theoretical analysis of their expressive power, temporal modeling capability boundaries, and the theoretical guarantee of alignment between time-series tokens and text tokens. There is no formal analysis of why cross-attention can better model temporal structure compared to soft prompting, and all conclusions about the advantages of the two architectures rely solely on experimental observations without theoretical support.
2. While the manuscript compares with tokenized, image-based, and text-only baselines (Section 4.2), it does not conduct direct quantitative performance comparisons with latest time-series LLM methods mentioned in the related work (e.g., ITFormer, InstructTime, Time2Lang). Only qualitative differences are described in the discussion section, which cannot fully demonstrate the performance superiority of OpenTSLM over existing SOTA methods in time-series reasoning tasks.
3. The related work section only briefly categorizes and lists existing time-series LLM methods, without in-depth analysis of the core limitations of prior works and how OpenTSLM specifically addresses these gaps. For cross-attention related work (SensorLM, Section 2.3), the manuscript only gives a one-paragraph introduction and simple difference description, without detailed comparison of core technical design differences and improvements.
4.  All experiments in the manuscript are limited to medical and healthcare time-series tasks, with no validation on other mainstream time-series domains such as finance, industrial monitoring, or general time-series forecasting. Although the conclusion mentions potential applications in other fields, there is no experimental evidence to prove that OpenTSLM is a general-purpose time-series language model framework, rather than a solution only adapted to medical time-series data.

---

> ### Author Rebuttal · Authors · 2026-03-30
>
> Thank you for the detailed and constructive feedback. We appreciate the suggestions to strengthen our manuscript.  However, we believe some parts around our work and novelty have been partially misunderstood.  We address questions and weaknesses below:
>
> ### Weaknesses
>
> > W1
>
> Please see Question 4.
>
> > W2
>
> Thank you for raising this. We added new evaluations of representative prior methods (ITFormer, ChatTS, ChatTime) on our datasets.
>
> |Model|HAR(F1/Acc)|Sleep(F1/Acc)|ECG(F1/Acc)|
> |-|-|-|-|
> |ITFormer|55.41/58.06|pending|pending|
> |ChatTS|57.63/64.05|41.31/73.23|8.91/42.32|
> |ChatTime|41.11/63.28|19.09/43.12|OOM|
> |**OpenTSLM**|**65.44/71.48**|**69.88/81.08**|**32.84/35.49**|
>
> OpenTSLM remains strongest performance across settings. Evaluations of InstructTime are ongoing. Time2Lang is not directly comparable, as it is limited to classification/regression with a fixed projection head.
>
> > W3
>
> We will further expand our comparison with prior work in the revision. Especially, regarding (i) **representation bottlenecks** from tokenization or softprompt approaches that scale poorly with long or multi-sensor time series, (ii) **task-specific designs** focused on classification or forecasting rather than open-ended reasoning (e.g., Time2Lang, MedualTime), and (iii) **limited use of pretrained LLM reasoning**, as some approaches train sensor-language models from scratch (e.g., SensorLM).
>
> SensorLM trains a sensor-language model from scratch for alignment and captioning, whereas OpenTSLM adapts pretrained LLMs for more general tasks. We will incorporate these clarifications in the revised manuscript.
>
> > W4
>
> See Question 3.
>
> ### Questions
>
> > Q1
>
> We use GPT-4o due to the lack of large-scale TSLM datasets. Crucially, it is not used as a predictor but as a _conditional generator_: given the ground-truth label and time-series (as plots), it explains observable temporal patterns. This reframes the task from inference to explanation, encouraging grounded descriptions. While GPT-4o performs poorly as a standalone predictor, it can still produce useful explanations when conditioned on the correct label, describing patterns rather than inferring them [2].
>
> To assess rationale quality, we iteratively performed manual inspection of n=20 rationales and tweaked our prompt if we found inconsistencies or misalignment with temporal patterns In the final setup (Appendix A2.1), all 20 sampled rationales showed no obvious inconsistencies, suggesting they are generally grounded. We acknowledge the lack of large-scale quantitative evaluation and will clarify this in the revision.
>
> Additionally, we conducted a structured expert evaluation on 84 ECG-QA samples with five cardiologists. OpenTSLM-Flamingo (LLaMA3.2-3B) achieved 92.9% correct or partially correct interpretations and 85.1% positive ratings for clinical context integration, indicating that the learned reasoning is largely meaningful.
>
> Our CoT dataset show that multimodal LLMs can learn to reason over raw time series. Training does not assume perfect rationales and has shown to tolerate some noise at scale. This is a first step, and future work will incorporate stronger validation (e.g., expert annotation, contrastive verification).
>
> > Q2
>
> See Weakness 2
>
> > Q3
>
> We agree that evaluating generalization beyond the medical domain is a natural next step and part of our ongoing work. In this paper, we intentionally focus on medical time series to avoid overextending claims while studying a domain with well-defined multimodal data and reasoning tasks.
>
> That said, we do not claim full cross-domain generalization and are actively exploring this direction in future work. We will further clarify this.
>
> > Q4
>
> SoftPrompt compresses the time series into a fixed token sequence concatenated with text, yielding a static representation whose effectiveness depends on the quality of this compression.
>
> In contrast, Flamingo maintains a separate time-series representation and uses gated cross-attention, enabling dynamic, query-dependent access across layers and better supporting multi-step reasoning over complex temporal dependencies.
>
> Tackling long sequences is a key motivation of our work. SoftPrompt scales poorly with sequence length: with $T$ text tokens, $M$ time-series tokens, and $R$ resampled latents, the attention cost per layer is $O((T+M)^2$ for SoftPrompt and $O(T^2 + T \cdot R)$ for Flamingo. Since $M$ grows with input length while $R$ remains fixed, Flamingo scales more favorably in compute and memory (see Appendix A.7).
>
> Accordingly, SoftPrompt is effective for shorter time series, whereas Flamingo is better suited for long or multivariate time-series requiring iterative reasoning. We note that both approaches learn alignment implicitly without formal guarantees. Establishing principled alignment remains an open problem, with SSL approaches explored in prior work [1] suggesting promising directions we plan to explore.
>
> 1. https://arxiv.org/abs/2506.09108
> 2. https://arxiv.org/abs/2409.11376

---

> > ### Author Rebuttal · Reviewer_XwpS · 2026-04-01
> >
> > Thank you for the detailed rebuttal. While the authors acknowledge several of my concerns, the majority of the responses rely on plans for future revisions rather than providing concrete evidence or analysis in the rebuttal itself. Thus, I will maintain my current score. Please refer to my detailed responses below.
> >
> > 1. The authors added partial SOTA method comparisons but have pending results for ITFormer on Sleep/ECG and incomplete evaluation for InstructTime, failing to fully demonstrate OpenTSLM's superiority across all tasks.
> >
> > 2. The authors confirm no cross-domain generalization experiments and only state it as future work, with no supplementary evidence for the generalizability of the OpenTSLM framework.
> >
> > 3. The theoretical analysis of the two architectures only supplements computational complexity, lacking formal analysis of expressive power and temporal modeling capability boundaries, and no theoretical guarantee for time-series-text token alignment is provided.

---

> > > ### Author Response · Authors · 2026-04-06
> > >
> > > We thank the reviewer again for the detailed feedback and for carefully considering our rebuttal. We appreciate the remaining concerns and would like to add additional results and clarifications below.
> > >
> > > > The authors added partial SOTA method comparisons but have pending results for ITFormer on Sleep/ECG and incomplete evaluation for InstructTime [...]
> > >
> > > Thank you for raising this point. We have now finished the evaluation of all requested models for comparison, on all datasets.
> > >
> > > The updated results are shown below:
> > >
> > > | Model        | HAR (F1/Acc) | Sleep (F1/Acc) | ECG (F1/Acc) |
> > > |--------------|-------------|---------------|-------------|
> > > | ITFormer     | 55.41 / 58.06 | 42.38 / 70.21 | 37.52 / 39.98 |
> > > | InstructTime | 61.73 / 65.07 | 42.57 / 59.31 | 20.71 / 25.16 |
> > > | ChatTS       | 57.63 / 64.05 | 41.31 / 73.23 | 8.91 / 42.32 |
> > > | ChatTime     | 41.11 / 63.28 | 19.09 / 43.12 | OOM |
> > > | **OpenTSLM** | **65.44 / 71.48** | **69.88 / 81.08** | **40.25 / 46.25** |
> > >
> > > OpenTSLM achieves the strongest performance across all models and all datasets. We note that ChatTime could not be trained on ECG-QA, as its tokenization-based approach leads to excessive sequence lengths and memory usage for this task (12-lead ECG, 30 seconds, 100Hz). We will include these complete comparisons in the revision and highlight important differences in architectures to account for challenges on different datasets (e.g., sequence length).
> > >
> > >
> > > > The authors confirm no cross-domain generalization experiments and only state it as future work, with no supplementary evidence for the generalizability of the OpenTSLM framework.
> > >
> > > We agree that cross-domain evaluation is an important direction and appreciate the reviewer’s point.
> > >
> > > In this work, we focused on an architectural contribution that enables scaling TSLMs to larger and more complex time-series inputs, supporting data of varying length from multiple sensors at once. Prior work predominantly relies on softprompt-based approaches, which we show do not scale well in VRAM with increasing sequence length and number of sensors. This may have limited their applicability to broader settings. Our results demonstrate that with the proposed cross-attention design, such scaling becomes feasible, and cross-domain generalization is now one clear next step for the field.
> > >
> > > At the same time, OpenTSLM is not specialized to the medical domain at the architectural level. The framework operates on generic multivariate time-series inputs, does not rely on domain-specific feature engineering, and directly leverages pretrained LLM reasoning capabilities, which supports its potential applicability beyond the evaluated setting.
> > >
> > > We will clarify in the revision claims around generlitzability and carefully make sure to not claim cross-domain generalization in this paper, and position this explicitly as an important next step potentially enabled by our approach.
> > >
> > >
> > > ---
> > >
> > > > The theoretical analysis of the two architectures only supplements computational complexity, lacking formal analysis of expressive power and temporal modeling capability boundaries, and no theoretical guarantee for time-series-text token alignment is provided.
> > >
> > > We thank the reviewer for this important point. We agree that a full formal characterization of expressive power, temporal modeling limits, and alignment guarantees is an important direction for future work.
> > >
> > > In this paper, our primary goal is to introduce and empirically validate the proposed architecture and its trade-offs compared to prior methods such as softprompting. Accordingly, we focus on empirically evaluating computational complexity and scaling behavior.
> > > While we do not provide formal guarantees, the empirical results suggest that the model is able to learn effective cross-modal alignment and temporal dependencies in practice. We will clarify this empirical scope more explicitly in the revision.
> > >
> > > We will revise the paper to explicitly acknowledge the lack of formal guarantees as a limitation, and highlight theoretical analysis of expressive power and alignment as an important direction for future research.
> > >
> > > Once again, thank you very much for your important suggestions and your assessment of our work. We believe they have contributed to strengthening the paper, and we are happy to include the additional experiments, clarifications and limitations in the revised manuscript.

---

### Official Review · Reviewer_USHB · 2026-03-09

**Soundness:** 3
**Presentation:** 3
**Significance:** 4
**Originality:** 3
**Overall Recommendation:** 5
**Confidence:** 3

**Summary:**

This paper proposes OpenTSLM, a family of Time-Series Language Models (TSLMs) that extend pretrained LLMs to natively process multivariate medical time-series together with text. The paper introduces two variants: OpenTSLM-SoftPrompt, which projects time-series embeddings into the LLM token space and interleaves them with text, and OpenTSLM-Flamingo, which treats time-series as a separate modality and integrates them through gated cross-attention in the style of Flamingo. The models are instantiated with open-weight LLM backbones and paired either with time-series encoders trained from scratch or with pretrained time-series foundation models such as Chronos-2.

The authors attempt to analyze a critical issue: current LLMs and multimodal LLMs remain poorly equipped to reason directly over raw time-series, especially in medical settings where temporal information is central. The manuscript's main area consists of multimodal reasoning over text and time-series, including both architectural design and benchmark construction. To support this, the paper also introduces three new chain-of-thought datasets—HAR-CoT, Sleep-CoT, and ECG-QA-CoT—and evaluates the models against tokenized-input baselines, vision-based approaches, and GPT-4o. The reported results suggest strong gains over text-only or tokenized approaches across tasks, and the paper further argues that the cross-attention variant scales better in VRAM as sequence count and length increase.

**Compliance With Llm Reviewing Policy:**

Affirmed.

**Key Questions For Authors:**

1.How much of the observed gain comes from better time-series representation versus better alignment to CoT-style supervision?

Since the training targets include generated rationales, it would help to better separate gains in temporal understanding from gains in producing benchmark-aligned explanations.

2.How robust are the conclusions across different rationale sources or supervision styles?

A brief discussion of whether the models remain strong when trained or evaluated with less templated reasoning targets would strengthen the paper.

3.How well does OpenTSLM transfer to more realistic clinical settings with noisy, irregular, or partially missing time-series?

The current experiments are convincing, but clarifying the expected boundary of applicability would improve the practical interpretation.

4.Can the authors more clearly quantify the trade-off between OpenTSLM-SoftPrompt and OpenTSLM-Flamingo beyond VRAM scaling?

The paper suggests that Flamingo scales better, but a concise summary of when one should prefer one variant over the other would be useful.

**Limitations:**

The paper is strong overall, but the limitations should be discussed more explicitly. In particular, the new CoT datasets depend substantially on LLM-generated rationales, which may entangle reasoning evaluation with explanation-style imitation. In addition, the evaluated settings appear more regular and curated than many real clinical time-series scenarios, where irregular sampling, missingness, and distribution shift are common. A clearer discussion of these boundaries would improve the paper.

**Strengths And Weaknesses:**

This is a strong and timely paper on an important multimodal extension of LLMs. The motivation is compelling, particularly for healthcare, where longitudinal sensor data are central but difficult to use with current language-centered interfaces. A major strength is that the paper does not stop at proposing one architecture: it compares two plausible integration strategies—soft prompting and explicit cross-attention—and studies both scratch-trained encoders and pretrained time-series foundation models. The benchmark contribution is also substantial. The introduction of three new reasoning-style datasets, especially ECG-QA-CoT with expert evaluation, meaningfully strengthens the paper’s practical relevance. Overall, the empirical story is quite persuasive: OpenTSLM consistently improves over tokenized baselines, preserves free-form reasoning, and provides useful scaling evidence for the Flamingo-style design.

My main concerns are about evaluation calibration and the extent to which “reasoning” is established rather than performance on CoT-formatted supervision. First, the new datasets appear to rely heavily on synthetic or LLM-generated rationales, which is understandable but raises the usual question of whether the models are learning genuine temporal reasoning or mostly reproducing target explanation style. Second, because the tasks are relatively benchmark-specific, it is still somewhat unclear how broadly the conclusions transfer to more open-ended or noisier real-world clinical time-series settings. Third, while the ECG expert study is a meaningful addition, the evidence presented in the main paper seems stronger for final-task performance than for rigorously disentangling reasoning quality from answer accuracy. These concerns do not outweigh the paper’s strengths, but they do suggest the claims should be scoped carefully.

---

> ### Author Rebuttal · Authors · 2026-03-30
>
> Thank you for your thoughtful review! We are glad you found the paper timely, technically strong, and impactful. We appreciate your recognition of both the architectural contributions and the new benchmarks. Please find our answers to your questions below.
>
> ## Questions
> > Q1
>
> We conducted an additional experiment comparing the Llama-1B baseline with both OpenTSLM variants under CoT supervision vs. label-only training (e.g., predicting “running” for HAR).
>
> | Model | HAR (F1) | Sleep (F1) | ECG-QA (F1) |
>  |-|-|-|-|
>  | Text-only CoT finetuned | 51.28 | 9.05 | OOM |
>  | OpenTSLM-SoftPrompt (CoT) | 65.44 | 69.88 | 32.84 |
>  | OpenTSLM-SoftPrompt (No CoT) | 60.29 | 64.90 | 29.92 |
>  | OpenTSLM-Flamingo (CoT) | 62.93 | 49.33 | 34.62 |
>  | OpenTSLM-Flamingo (No CoT) | 59.99 | 48.24 | 31.31 |
>
> Both variants significantly outperform the text-only baseline, indicating gains are mainly due to improved time-series representations rather than CoT. Removing CoT slightly reduces performance, but models remain competitive. We find that label-only training models produce more repetitive outputs, while CoT improves output quality.
>
> > Q2
>
> Thank you for raising this point. We agree that evaluating robustness across diverse rationale sources and supervision styles is an important next step to determine whether the model learns true temporal reasoning rather than overfitting to specific explanation formats. Our CoT vs. label-only ablation provides an initial signal by isolating rationale supervision, showing that most performance gains persist even without explicit reasoning targets, suggesting the model may not be reliant on a specific CoT style.
>
> Our current datasets rely on templated or LLM-generated rationales, which may introduce stylistic bias. Expanding to diverse sources (e.g., human-authored) is important but challenging due to (1) limited availability and high cost of high-quality annotations and (2) difficulty isolating reasoning ability from confounding factors when rationale styles vary (varying rationale styles may alter task difficulty, information content). We are curating larger-scale datasets to enable such analysis in future work and will clarify this in the revision.
>
> > Q3
>
> We have added additional robustness experiments under controlled noise and sensor failure scenarios.
> 1.  **Additive Noise** We add increasing levels of Gaussian noise to the inputs. Noise is generated with the same distribution as the normalized signal and linearly interpolated with the original signal:
> $\tilde{x} = (1 - \alpha)\, x + \alpha\, \epsilon, \quad \epsilon \sim \mathcal{N}(0, \sigma^2) $
> where $x$ is the signal, $\tilde{x}$ the noisy variant, $\alpha=0.1$, and $\epsilon$ Gaussian noise.
>
> 2. **Sensor failure** Instead of adding noise, we replace all sensors completely with noise.
>
> |Noise-level|TSQA (F1)|HAR (F1)|Sleep (F1)|ECG (F1)|
> |---|---|---|---|---|
> |Normal|94.08|62.93|49.33|41.58|
> |10%|91.55|62.91|43.74|40.27|
> |30%|77.11|55.56|31.27|39.80|
> |50%|50.82|43.10|12.76|39.92|
> |70%|35.28|34.15|12.46|39.46|
> | Sensor Failure|32.48|10.59| 16.50| 28.31|
>
> Performance degrades under noise, but the model does not collapse and still produces outputs. The predictions may reflect priors rather than grounded reasoning when signals are fully corrupted.
>
>   > Q4
>
> Thank you for highlighting this point! SoftPrompt and Flamingo variants differ fundamentally in how time-series information is exposed to the LLM. In SoftPrompt, the time-series encoder generates embeddings that are concatenated with text embeddings and processed via self-attention. This compresses the time-series into a tokenized prefix/infix, creating a static representation for subsequent language reasoning.
>
> Flamingo keeps the time-series as a separate latent memory. Via gated cross-attention, text tokens can query this memory across decoding layers. This supports dynamic access to different aspects of the time-series during reasoning, rather than relying on a single compressed representation.
>
> Computationally, let T be text tokens, M time-series tokens in SoftPrompt, and R resampled latents in Flamingo. The attention cost is $O((T+M)^2)$ for SoftPrompt and $O(T^2 + T \cdot R)$ for Flamingo. Since M grows with input length while R can be fixed, Flamingo scales better for long or multiple time-series in both compute and memory.
>
> We recommend SoftPrompt for short time-series tasks that rely on compact summaries within the LLM context, offering strong performance with minimal fine-tuning (LoRA) and good VRAM efficiency, though its memory scales with sequence length. Flamingo, in contrast, enables dynamic, multi-step reasoning over temporal signals via cross-attention, making it preferable for longer or multivariate time-series (e.g., ECG, IMU), where it provides near-constant memory usage and stronger performance on complex datasets.
>
> ## Limitations
> Thank you for pointing out additional discussion points for the limitations. We will incorporate them in the revised version!

---

> > ### Author Rebuttal · Reviewer_USHB · 2026-04-04
> >
> > Thank you to the authors for their detailed rebuttal and clarifications. I have carefully read the response and appreciate the additional explanations provided. After considering the rebuttal, I believe my main concerns have been adequately addressed to the extent possible, but my overall assessment of the paper remains unchanged. Therefore, I decide to maintain my original score.

---

> > > ### Author Response · Authors · 2026-04-06
> > >
> > > Thank you very much for your thoughtful review and for carefully engaging with our rebuttal. We sincerely appreciate your insightful comments and constructive feedback throughout the process. We believe your suggestions have helped strengthen the paper, and we will incorporate them to further improve clarity, positioning, and discussion of limitations in the final version.
> > >
> > > Thank you once again for your time, your positive assessment of our work, and your valuable comments.

---

### Official Review · Reviewer_Zo5j · 2026-03-10

**Soundness:** 2
**Presentation:** 3
**Significance:** 2
**Originality:** 3
**Overall Recommendation:** 4
**Confidence:** 4

**Summary:**

This paper proposes OpenTSLM, a framework that enables large language models (LLMs) to process and reason about time-series data together with text. The authors explore two integration strategies: soft prompting and cross-attention. They also introduce three new datasets (HAR-CoT, Sleep-CoT, and ECG-QA-CoT) for time-series reasoning tasks with natural language outputs. Experiments evaluate the proposed models across multiple LLM backbones and domain-specific datasets, showing that the cross-attention strategy is both more memory efficient and achieves better performance.

**Compliance With Llm Reviewing Policy:**

Affirmed.

**Final Justification:**

The rebuttal addressed several of my concerns and improved the paper’s clarity, and I increased my assessment of originality. However, I still have two concerns. I appreciate the new joint training results across datasets, which help address my concern and show that unified training can improve performance. However, these results make cross-domain training and evaluation even more aligned with the paper’s framing than the current dataset-specific setting in the submission. Given the title and positioning, I still think this aspect should be emphasized and discussed as part of the main results. I also still have some reservations about the reliability of GPT-4o-generated rationales as supervision, given its poor zero-shot prediction performance. Overall, I have increased my score to weak accept.

**Key Questions For Authors:**

Please see weaknesses.

**Limitations:**

yes

**Strengths And Weaknesses:**

**Strengths**

S1. The paper introduces three new datasets (HAR-CoT, Sleep-CoT, and ECG-QA-CoT) for reasoning over time-series signals with natural language outputs. These datasets may be useful resources for research on time-series reasoning and multimodal LLMs, although the CoT are synthesized by GPT-4o.

S2. Clear problem formulation and presentation. Enabling LLMs to reason over time-series data is a relevant and emerging direction, particularly for domains such as healthcare where temporal signals are common.

S3. The paper studies two integration strategies (soft prompting and cross-attention) and evaluates them across multiple model backbones and datasets.



****
**Weaknesses**

W1. Limited comparison with existing time-series-to-text language models.
While the paper compares different architectures and training mechanisms within the proposed framework, it does not evaluate or fine-tune a broader set of existing time-series language models as baselines. As a result, it is unclear how the proposed methods perform relative to prior time-series LLM approaches when those models are adapted to the same datasets and tasks. A more comprehensive comparison would help clarify the contribution of the proposed architecture within the broader literature.
- The abundant time series language models adopt varying architectures. The paper does a good job summarizing them in the related work (SensorLLM, ITFormer, InstrucTime, ChatTS, Time2Lang, MedTsLLM, MedualTime). Another relevant work is “ChatTime: A Unified Multimodal Time Series Foundation Model Bridging Numerical and Textual Data.” However, without adapting these prior methods to the same datasets and tasks, we cannot clearly determine whether the performances by OpenTSLM comes from a better architecture or simply from differences in the dataset or training setup.
- Such comparisons are especially important for time-series to text generation models (e.g., ChatTS, ChatTime), which appear closely related to the proposed approach.


W2. Train, validation, and test come from the same data distribution. The reported performances appear to reflect domain- or dataset-specific performance rather than true generalization capability. Did the authors consider conducting cross-dataset validation experiments? If not, the paper title is somewhat misleading, as the title appears to propose general-purpose or foundation time-series language models, while the evaluation only demonstrates performance within individual datasets.


W3. Limited methodological novelty. Many components of the proposed framework appear to rely on existing techniques in the literature, such as time-series encoders, soft prompting, and cross-attention mechanisms for multimodal fusion. While the paper integrates these components into a unified framework, the methodological contributions seem largely incremental. As a result, the level of architectural or algorithmic novelty relative to prior time-series language modeling works may be limited.

W4. Experiments missing ablations and need clarifications.
- For the patch encoder design, did the authors ablate performance when the patch size p varies? How should one determine the optimal design for a given dataset?
- Did the authors conduct an ablation study without the stage-one synthetic time-series pretraining? It would be helpful to understand how much the final performance depends on this synthetic pretraining stage.
- The reasoning (CoT) dataset is generated synthetically using GPT-4o. However, in lines 324–328 the paper shows that GPT-4o itself performs poorly on the time-series classification tasks. At the same time, the authors claim that “even frontier LLMs like GPT-4o are poorly suited for time-series reasoning and that time-series must be treated as a distinct modality.” If GPT-4o struggles with these tasks, it is unclear whether the generated rationales accurately reflect correct reasoning over time-series signals. Could the authors elaborate on the rationale for using GPT-4o to generate the reasoning supervision despite its weak performance on these tasks?
- Several LLM baseline models achieve ~0.0 F1 performance because they fail to produce the expected Answer: {answer} template or instead repeat the input prompt. This raises the possibility that the evaluation may partially reflect formatting compliance rather than task capability. It would be helpful to clarify whether alternative parsing strategies or prompt adjustments were considered to ensure a fair comparison with baseline models. Did the authors use robust answer parsing? Can classification accuracy be evaluated independently of output format? Did the authors try few-shot prompting to enforce the answer format?

---

> ### Author Rebuttal · Authors · 2026-03-30
>
> We thank the reviewer for the detailed and constructive feedback. We address each point below:
>
> > W1
>
> We agree that a comparison with existing time-series-to-text models are important to better position our contribution. We have added additional experiments with representative prior methods (ITFormer, ChatTS, ChatTime) on our datasets and tasks:
>
> |Model|HAR-CoT(F1/Acc)|Sleep-CoT(F1/Acc)|ECG-QA-CoT(F1/Acc)|
> |-|-|-|-|
> |ITFormer|55.41/58.06|pending|pending|
> |ChatTS|57.63/64.05|41.31/73.23|8.91/42.32|
> |ChatTime|41.11/63.28|19.09/43.12|OOM|
> |**OpenTSLM**|**65.44/71.48**|**69.88/81.08**|**32.84/35.49**|
>
> These results show that prior methods do not transfer consistently across tasks and show scaling issues (e.g., ChatTime OOM). The better performance and stability is indicating gains beyond dataset or training differences. We will include these comparisons in the final manuscript.
>
> > W2
>
> Thank you for raising this point. We agree that our current evaluation mainly reflects in-domain performance, as each dataset is split from the same distribution. However, our evaluation spans multiple heterogeneous datasets and tasks, HAR, Sleep-CoT, and ECG-QA-CoT, which differ in modality, label space, and reasoning requirements, partially probing cross-domain robustness. We agree that explicit cross-dataset validation would provide a stronger assessment, but this is non-trivial due to (i) differing label spaces and task formulations, (ii) heterogeneous sensor modalities, and (iii) the lack of large benchmarks for multimodal time-series reasoning. We will clarify this limitation in the revision and moderate claims on generalization. We view cross-dataset and zero-shot generalization as important directions for future work.
>
> > W3
>
> While OpenTSLM builds on multimodal fusion mechanisms, our contribution is not a direct transfer of these mechanisms, but a systematic redesign of how temporal data integrates into LLMs. Time-series data differs from vision: it is sequential (variable length), irregular, and has long-range dependencies, making integration and scaling nontrivial.
> Although soft prompting and cross-attention exist in prior work, their (scaling) behavior for time-series reasoning remain unestablished. Most prior TSLMs use soft prompting, which captures temporal structure only implicitly. We show this becomes inefficient as sequence length increases.
> Our key contribution is to **decouple temporal modeling from language reasoning** via explicit modality separation and gated cross-attention, enabling scalable multivariate time-series reasoning. We also provide the first systematic comparison (to our knowledge) of soft prompting vs cross-attention for time-series, showing clear performance and memory trade-offs.
> Thus, our novelty lies in resolving the design space for time-series–LLM integration, which is essential for scaling TSLMs to real-world settings.
>
> > W4
>
> We thank the reviewer for these suggestions.
>
> **Patch size.** We ablated $p \in \{1,2,4,8\}$ using LLaMA3.2-1B.
>
> |Model|HARF1|SleepF1|ECGF1|
> |-|-|-|-|
> |SoftPrompt p1|57.78|59.34|29.92|
> |SoftPrompt p2|64.29|60.65|31.29|
> |SoftPrompt p4|**65.44**|**69.88**|32.84|
> |SoftPrompt p8|64.97|69.74|33.56|
> |Flamingo p1|60.13|45.44|30.01|
> |Flamingo p2|62.22|48.63|32.47|
> |Flamingo p4|62.93|49.33|34.62|
> |Flamingo p8|62.44|49.41|**35.74**|
>
> Performance peaks at $p=4$ (with $p=8$ comparable), while smaller patches underperform. We will include the full results table in the final manuscript.
>
> **Synthetic stage-one pretraining.** We thank the reviewer for this suggestion. We are currently running this ablation; results are pending and will be included in the final manuscript.
>
> **GPT-4o-generated rationales.** We acknowledge the concern regarding rationale quality. GPT-4o is not used as a predictor here, but as a *conditional generator*: it is given the ground-truth label together with the time-series visualization, and asked to explain observable temporal patterns. This substantially reduces the inference burden compared to solving the task directly.
>
> To assess rationale quality, we additionally performed expert evaluation on 84 ECG-QA samples with five cardiologists (see Section 4.5). OpenTSLM Flamingo LLaMA3.2-3B achieved 92.9% correct or partially correct interpretations, with 85.1% positive ratings for clinical context integration. This suggests that the generated supervision is sufficiently meaningful despite GPT-4o’s poor zero-shot time-series performance.
>
> **Formatting vs task capability.** We tested relaxed parsing (e.g., keyword matching) and observed no material change. Manual inspection shows baselines rarely produce correct answers in any format, instead repeating prompts or generating irrelevant outputs. Fine-tuning improves format compliance but not accuracy, indicating the gap is due to difficulty processing tokenized time-series, not formatting.
> We will incorporate these comparisons, ablations, and clarifications in the revised version.

---

> > ### Author Rebuttal · Reviewer_Zo5j · 2026-04-02
> >
> > Thank you for the rebuttal and many additional experiments. Many of my concerns were addressed, I would like to increase the score of originality. However, I still have several concerns.
> >
> > 1. The evidence for cross-dataset generalization remains limited. This matters because the paper’s title and framing suggest “an open-source framework and family of time series language models,” where broader cross-domain pretraining and generalizability would be expected. However, the current experiments appear to train a separate model for each dataset or task. For example, the ECG/Sleep model needs to be separately trained. Have the authors considered training one unified model jointly on all datasets to better test cross-domain generality?
> >
> > 2. The pending ablation without stage-one synthetic pretraining leaves a methodological question unresolved.
> >
> > 3. (minor) Thank you for the human evaluation on 84 ECG-QA samples. I still have some reservations about the quality of the GPT-4o-generated rationales given its poor zero-shot prediction performance. As a ‘conditional generator’, GPT-4o still relies on autoregressive token generation. Although rationale generation is easier with the ground-truth label provided, the reliability of these rationales (based on time-series) by GPT-4o remains unclear given its poor zero-shot prediction (based on time series).

---

> > > ### Author Response · Authors · 2026-04-06
> > >
> > > Thank you for your positive feedback and for raising these important remaining concerns. We are glad that many of the earlier points were resolved, and we appreciate the opportunity to further strengthen the paper. We have done additional experiments to address your remaining points below.
> > >
> > >
> > > >  The evidence for cross-dataset generalization remains limited. [...]
> > >
> > > Thank you for raising this very important point.  To address this, we conducted additional experiments where we train OpenTSLM-Flamingo Llama-3B jointly across all datasets, instead of training separate models per task. Due to time constraints during the rebuttal period, we focused on training only the largest model (3B), as we believe higher capacity is necessary to effectively learn shared representations across heterogeneous domains. Please find our results in the table below.
> > >
> > > | Model                      | TSQA  | HAR-CoT | Sleep-CoT | ECG-QA-CoT |
> > > |-|-|-|-|-|
> > > | Llama3.2-3B (OpenTSLM)    | 90.14 | 62.77   | 45.45     | 40.25      |
> > > | Llama3.2-3B (Merged)      | 92.73 | 69.31   | 52.13     | 41.55      |
> > > | Improvements merged vs. single    | +2.59  | +6.54     | +6.68       | +1.30        |
> > >
> > > Importantly, our setup does not follow a classical multi-task learning formulation with separate task-specific objectives. Instead, we train a single causal generative model that maps time-series inputs to textual outputs (labels and rationales) across all datasets. From this perspective, joint training can be viewed as increasing the diversity of supervision rather than introducing explicit task decomposition.
> > > While the observed improvements are consistent with improved cross-domain generalization, we acknowledge that they may also be partially explained by implicit regularization effects due to increased data diversity. We are not able to fully disentangle these factors at this stage but we will highlight and elaborateon this in the discussion of the revision manuscript. Additionally, the imbalance in dataset sizes may bias training toward larger datasets and influence the observed performance (please see Appendix A.1).
> > > Together, these limitations point to two important directions for future work: (i) more controlled studies to isolate true cross-domain generalization effects, and (ii) systematically investigating the impact of dataset balance when training larger TSLMs.
> > >
> > >
> > >
> > > >  The pending ablation without stage-one synthetic pretraining leaves a methodological question unresolved.
> > >
> > > We thank the reviewer for highlighting this point. We have now completed the requested ablation, using OpenTSLM-Flamingo on the Llama-1B model.
> > >
> > > | Model                       | HAR-CoT | Sleep-CoT | ECG-QA-CoT |
> > > |---------------------------|---------|-----------|------------|
> > > | OpenTSLM-Flamingo/Llama3.2-1B (original)    | 62.93   | 49.33     | 34.62      |
> > > | OpenTSLM-Flamingo/Llama3.2-1B (without synthetic pretraining)    | 60.28   | 42.59     | 34.03      |
> > >
> > > Removing synthetic pretraining leads to worse performance, particularly on tasks with limited data availability such as Sleep-CoT (please refer to Section A.1 for an overview of dataset sizes). This indicates that synthetic pretraining provides a beneficial initialization, especially for tasks requiring structured reasoning or involving noisier signals. We will include this ablation and corresponding discussion in the revised version.
> > >
> > > >(minor) Thank you for the human evaluation on 84 ECG-QA samples. I still have some reservations about the quality of the GPT-4o-generated rationales given its poor zero-shot prediction performance.
> > >
> > > Thank you very much for this important point. We fully agree that, ideally, rationales should be curated or verified by domain experts to ensure faithful grounding in the underlying time-series signals. However, this reflects a broader and currently unresolved challenge in the field of time-series language models, where large-scale, expert-annotated datasets with high-quality rationales are extremely scarce. Curating large datasets is very time-intensive and requires expert annotators, esp. in the medical domain. As a result, many existing works rely on weak supervision signals such as LLM-generated rationales.
> > > We acknowledge that such rationales may not be guaranteed be fully reliable or signal-grounded, and we will clarify this limitation more explicitly in the revised paper. At the same time, we are actively working toward curating larger, expert-informed datasets to improve the quality and reliability of supervision in future work.

---

### Official Review · Reviewer_128K · 2026-03-11

**Soundness:** 3
**Presentation:** 3
**Significance:** 3
**Originality:** 2
**Overall Recommendation:** 4
**Confidence:** 3

**Summary:**

This paper presents OpenTSLM, a framework aimed at integrating multivariate time series data as "native modalities" into large language models (LLMs). In response to the demand for the collaborative analysis of long-term, multivariable physiological signals and text reports in medical scenarios, the authors have designed a dedicated time series Tokenizer and alignment mechanism. In addition, the author highlighted two architectures: OpenTSLM-SoftPrompt (Lightweight insertion) and OpenTSLM-Flamingo (Deep cross-modal attention), and conducted experiments on multiple public medical datasets.

**Compliance With Llm Reviewing Policy:**

Affirmed.

**Final Justification:**

The authors have addressed my concerns, so I raise my score to 4.

**Key Questions For Authors:**

1. When using the Llama model as the backbone, the transition from 1B to 3B brought about a significant increase in VRAM memory. However, in the experimental results, it mostly decreased. Why is that?
2. Does OpenTSLM merely enhance the model's acceptance ability through post-training? Does this imply that it requires a higher computational cost than other TimesLLM methods？
3. Without the assistance of text information, would LLMS still outperform specialized small temporal models in temporal tasks solely based on temporal modalities?

**Limitations:**

yes

**Strengths And Weaknesses:**

Strengths:
1. Based on the existing large models, by integrating the time series as a native modality into the pre-trained large language model, the model can reason and understand multiple time series through **natural language cues (prompting)**.
2. Three Chain-of-Thought (CoT) datasets were constructed, providing a basis for the joint reasoning task of medical texts and time series.
3. The author made a detailed comparison of two mainstream modal integration paradigms: SoftPrompt (lightweight) and Flamingo style (heavy), and analyzed their respective advantages, disadvantages, and trade-offs.
4. Compared with the native LLMs, the proposed OpenTSLM has achieved significant performance improvements.

Weakness:
1. OpenTSLM's core architecture (SoftPrompt and the Flamingo structure based on Gated Cross-Attention) has been very mature in the multimodal field, especially in the vision-language field. This article seems to have appropriated this architecture onto time series, but it appears somewhat thin in the unique design of this specific mode of "time series", as if it were a success in application rather than an innovation in architecture.
2. The currently adopted Learned Tokenizer performs well on specific datasets, but its generalization ability is questionable for unseen sampling frequencies or sensor data with completely different dimensions.
3. For OpenTSLM, if the input time series contains sensor noise or lead detachment, will the model produce seriously misleading descriptions? The text lacks Robustness tests for such abnormal situations.
4. Unlike text or images, medical physiological signals (such as 24-hour ECG) usually contain extremely long sequences. Although the paper mentioned block processing, it did not analyze in detail the computational costs and bottlenecks brought about by the increase in sequence length.

---

> ### Author Rebuttal · Authors · 2026-03-30
>
> Thank you for your constructive feedback! Below, we would like to clarify several points where our novelty and contributions may have been misunderstood. We also added new experiments to specifically evaluate robustness under noise.
>
> ## Weaknesses
>
> > W1
>
> Thank you for this insightful comment. While OpenTSLM builds on multimodal fusion mechanisms, our contribution is not a direct transfer of these mechanisms, but a systematic redesign of how temporal data integrates into LLMs. Time-series data differs from vision: it is sequential (variable length), irregular, and has long-range dependencies, making integration and scaling nontrivial.
>
> Although soft prompting and cross-attention exist in prior work, their (scaling) behavior for time-series reasoning remain unestablished. Most prior TSLMs use soft prompting [1,2] which captures temporal structure only implicitly. We show this becomes inefficient as sequence length increases.
>
> Our key contribution is **decoupling temporal modeling from language reasoning** via gated cross-attention, enabling scalable reasoning. Additionally, we provide the **first systematic comparison** (to our knowledge) of implicit (soft prompting) vs. explicit (cross-attention) fusion for time-series, revealing important performance and memory trade-offs. Thus, our novelty lies  in **resolving the design space for time-series–LLM integration**, which is essential for scaling TSLMs to real-world settings.
>
> > W2
>
> We agree that generalization of learned time-series tokenizers to unseen settings is a key challenge. Our current tokenizer is trained jointly on task-specific datasets and may have limited out-of-distribution generalization. However, OpenTSLM is tokenizer-agnostic and supports aligning pretrained time-series foundation models (TSFMs) such as Chronos-2, trained on large, diverse datasets with better generalization. Improving robustness remains important future work, with promising directions including aligning pretrained encoders, multi-resolution training, and self-supervised learning. A major limitation is the scarcity of large, heterogeneous time-series datasets, which we will clarify in the revised text.
>
> > W3
>
> We added experiments evaluating robustness to noise (Flamingo-Llama-1B):
> $\tilde{x}=(1-\alpha)x+\alpha\epsilon$,
> where $x$ is the signal, $\tilde{x}$ the noisy variant, $\alpha=0.1$, and $\epsilon$ Gaussian noise.
>
> |Noise|TSQA (F1)|HAR (F1)|Sleep (F1)|ECG (F1)|
> |---|---|---|---|---|
> |Normal|94.08|62.93|49.33|41.58|
> |10%|91.55|62.91|43.74|40.27|
> |30%|77.11|55.56|31.27|39.80|
> |50%|50.82|43.10|12.76|39.92|
> |70%|35.28|34.15|12.46|39.46|
> |100%|32.48|20.59|12.50|28.31|
>
> Performance degrades under noise, but the model does not collapse and still produces outputs. The predictions may reflect priors rather than grounded reasoning when signals are fully corrupted.
>
> > W4
>
> Thank you for raising this point. Tackling long physiological sequences is a key motivation of our work. We show that soft prompting scales poorly with sequence length.  Let T be text tokens, M time-series tokens, and R resampled latents (Flamingo). Attention cost per layer is $O((T+M)^2)$ for SoftPrompt and $O(T^2 + T \cdot R)$ for Flamingo. Since M grows with input length but R typically remains fixed, Flamingo scales better in compute and memory. We analyze this in Appendix A.7. Recent literature also compares OpenTSLM-Flamingo vs. ITFormer on long-context retrieval tasks [3].
>
> ## Questions
> > Q1
>
> The 3B model may overfit to limited data and memorize few-shot patterns, while the 1B model learns more generalizable representations. Larger models also have stronger pretrained priors, making adaptation to numerical time-series harder under limited supervision.
>
> >  Q2
>
> Thank you for this question. OpenTSLM is not a post-training approach, but turns pretrained LLMs into multimodal models by integrating time-series as a native modality via new components. CoT supervision is used only for output alignment. We highlight that the OpenTSLM-Flamingo variant is in fact more efficient than prior TimeLLM approaches, especially for long or multiple time-series. Prior methods rely on tokenization or softprompt, which increase token count and lead to quadratic scaling. See W4 for complexity discussion.
>
> > Q3
> >
> Specialized models may outperform on narrowly defined tasks such as classification or forecasting, but this is not a fully fair comparison because the two serve different goals. Task-specific models are optimized for fixed settings, such as fixed sensors, sampling rates, input lengths, and label spaces. In contrast, OpenTSLM enables multimodal natural-language reasoning over time-series. Our goal is to extend LLMs with temporal reasoning, supporting explanations, external context, and flexible handling of varying sensors and tasks, analogous to how vision-language models complement specialized vision models.
>
> 1. https://arxiv.org/abs/2412.03104
> 2. https://arxiv.org/abs/2412.11376
> 3. https://arxiv.org/abs/2602.14200

---

> > ### Author Rebuttal · Reviewer_128K · 2026-04-02
> >
> > Thanks for your reply. However, I still have several concerns. It seems that the scaling rule of traditional LLMs has failed (even for models as large as 270M, some experimental performances are the best, Table 3). Does this mean that temporal understanding itself does not require a very large model as a foundation to support it? I think the author's discussion of this phenomenon is not deep enough. If some relevant analysis and experiments could be provided, **I would increase my score. Since one of the main contributions of this paper is the systematic analysis of sequential large models.**

---

> > > ### Author Response · Authors · 2026-04-06
> > >
> > > Thank you very much for your insightful comment. We appreciate the opportunity to clarify this important point.
> > >
> > > First, please note that in our experiments, we train models of different sizes (270M, 1B, 3B) on **the exact same amount of data**. The observation that smaller models can outperform larger models on the same amount of (limited) data does not contradict scaling laws. Scaling laws predict **improved performance when both model size and data scale increase** [1]. When data is insufficient, **larger models tend to be undertrained and cannot fully utilize their higher capacity**, which can lead to inferior performance compared to smaller models.
> > >
> > > To further investigate this, we conducted additional experiments on the ECG-QA dataset by varying the amount of available training data, using 10%, 25%, 50%, 75%, and 100% of all available data, while keeping the test set fixed. This means that during training **we only include a subset of all available data, but still evaluate on the same test dataset** as before.
> > > The results are summarized below (**please mind the link to a plot below, which shows the scaling more clearly**):
> > > | Used training data (%) | Gemma-3 270M (F1) | Llama-1B (F1) | Llama-3B (F1) |
> > > |------------------|------------------|--------------|--------------|
> > > | 10               | 5.75             | 2.46         | 1.06         |
> > > | 25               | 11.41            | 18.67        | 13.47        |
> > > | 50               | 28.23            | 31.42        | 26.40        |
> > > | 75               | 31.91            | 33.30        | 34.00        |
> > > | 100              | 32.71            | 34.62        | 40.25        |
> > >
> > > # [[Click] View plot showing scaling performance](https://postimg.cc/FdJDS4dD)
> > >
> > >
> > >
> > > These results clearly demonstrate a data-and-model-size-dependent scaling behavior. In **low-data regimes (e.g., 10%), smaller models perform better**, suggesting that larger models are undertrained and cannot effectively utilize their capacity. **As the amount of training data increases, the performance of larger models improves further while smaller models show diminishing performance gains.**
> > >
> > > Notably, at higher data fractions (75% and 100%), the 3B model surpasses both the 270M and 1B models, achieving the strongest overall performance, which aligns with established scaling laws indicating that larger models require sufficient data to fully realize their capacity [1]. Taken together, these additional experiments reinforce that our original observations are best understood as a consequence of operating in a data-limited regime, rather than reflecting any deviation from expected scaling behavior. We will incorporate this analysis and discussion into the revised manuscript to improve clarity and strengthen the paper.
> > >
> > > We hope that this additional analysis sufficiently addresses your concerns and clarifies the observed phenomenon. If you find this clarification and the newly provided experimental evidence satisfactory, we would kindly ask you to consider raising your score.
> > >
> > > [1] Kaplan, J., McCandlish, S., Henighan, T., Brown, T. B., Chess, B., Child, R., Gray, S., Radford, A., Wu, J. & Amodei, D. (2020). Scaling Laws for Neural Language Models.. CoRR, abs/2001.08361.

---

### Decision · Program_Chairs · 2026-04-30

**Decision:**

Accept (regular)

**Comment:**

The submission proposes methodology for incorporating multi-modal time series data with LLMs in a clinical application.  Despite some concerns, many of which were satisfactorily addressed by the authors, all reviewers agree the submission should be accepted, and I concur.